# Sex-specific and pleiotropic effects underlying kidney function identified from GWAS meta-analysis

Sarah E. Graham [1], Jonas B. Nielsen [1], Matthew Zawistowski [2], Wei Zhou[3], Lars G. Fritsche [2], Maiken E. Gabrielsen[4,5], Anne Heidi Skogholt[4,5,6], Ida Surakka[1], Whitney E. Hornsby[1], Damian Fermin[7], Daniel B. Larach[8], Sachin Kheterpal[8], Chad M. Brummett[8], Seunggeun Lee [2], Hyun Min Kang[2], Goncalo R. Abecasis[2], Solfrid Romundstad[6,9], Stein Hallan[6,10], Matthew G. Sampson [7], Kristian Hveem[4,5,11] & Cristen J. Willer [1,3,12]

Chronic kidney disease (CKD) is a growing health burden currently affecting 10–15% of adults worldwide. Estimated glomerular filtration rate (eGFR) as a marker of kidney function is commonly used to diagnose CKD. We analyze eGFR data from the Nord-Trøndelag Health Study and Michigan Genomics Initiative and perform a GWAS meta-analysis with public summary statistics, more than doubling the sample size of previous meta-analyses. We identify 147 loci (53 novel) associated with eGFR, including genes involved in transcriptional regulation, kidney development, cellular signaling, metabolism, and solute transport. Additionally, sex-stratified analysis identifies one locus with more significant effects in women than men. Using genetic risk scores constructed from these eGFR meta-analysis results, we show that associated variants are generally predictive of CKD with only modest improvements in detection compared with other known clinical risk factors. Collectively, these results yield additional insight into the genetic factors underlying kidney function and progression to CKD.

[1] Department of Internal Medicine: Cardiology, University of Michigan, Ann Arbor 48109 MI, USA. [2] Department of Biostatistics: Center for Statistical Genetics, University of Michigan, Ann Arbor 48109 MI, USA. [3] Department of Computational Medicine and Bioinformatics, University of Michigan, Ann Arbor 48109 MI, USA. [4] K.G. Jebsen Center for Genetic Epidemiology, Faculty of Medicine and Health Sciences, Norwegian University of Science and Technology, Trondheim 7491, Norway. [5] Department of Public Health and Nursing, Faculty of Medicine and Health Sciences, Norwegian University of Science and Technology, Trondheim 7491, Norway. [6] Department of Clinical and Molecular Medicine, Faculty of Medicine and Health Sciences, Norwegian University of Science and Technology, Trondheim 7491, Norway. [7] Department of Pediatrics: Pediatric Nephrology, University of Michigan, Ann Arbor 48109 MI, USA. [8] Department of Anesthesiology, University of Michigan, Ann Arbor 48109 MI, USA. [9] Department of Internal Medicine, Levanger Hospital, Health Trust Nord-Trøndelag, Levanger 7600, Norway. [10] Department of Nephrology, St Olav Hospital, Trondheim 7491, Norway. [11] HUNT Research Centre, Department of Public Health and General Practice, Norwegian University of Science and Technology, Levanger 7600, Norway. [12] Department of Human Genetics, University of Michigan, Ann Arbor 48109 MI, USA. Correspondence and requests for materials should be addressed to K.H. (email: kristian.hveem@ntnu.no) or to C.J.W. (email: cristen@umich.edu)

Chronic kidney disease (CKD) is a common condition affecting ~11% of adults in Norway and ~15% in the United States[1,2]. Due to specific comorbidities (namely diabetes) and an aging population, CKD is expected to continue to rise in global prevalence[3]. Estimated glomerular filtration rate (eGFR) provides an assessment of kidney function and it is estimated based on serum creatinine levels with adjustment for age, race, and sex. eGFR levels below 60 mL/min/1.73 m$^2$ generally characterize chronic kidney disease[4], with varying severity classified by both albuminuria and eGFR levels. A subset of individuals with CKD have accelerated renal function decline and progress to end stage renal disease (ESRD).

Several other health conditions affect kidney function. Chronic diseases such as diabetes and hypertension directly influence the development of CKD, with environmental factors such as smoking accelerating disease progression[5]. Advanced stages of CKD/ESRD necessitate dialysis or transplantation and are associated with an increased risk of cardiovascular disease and death[6].

It has been estimated that about one-third of the variation in eGFR levels can be attributed to genetic factors[7], with the remaining variability due to environmental effects. Previous genome-wide association studies (GWAS) and meta-analyses have identified a number of loci associated with serum creatinine, eGFR, or CKD[8–18]. However, the introduction of denser imputation panels, including the Haplotype Reference Consortium[19] (HRC), and the recent rise in large-scale biobanks has enabled larger sample sizes and a greater number of variants than previously studied. Analysis of these new and more densely imputed datasets are expected to identify genetic regions influencing these traits not previously found[20].

We analyze samples from the Michigan Genomics Initiative (MGI) and the Nord-Trøndelag Health Study (HUNT), and impute using HRC and a combined HRC and ancestry-specific panel, respectively, for association with eGFR. Finally, we perform a meta-analysis of eGFR associations with two other cohorts to uncover additional genetic variants contributing to kidney function. We identify 147 loci associated with eGFR, including 53 novel loci and one locus with a significantly larger effect in women than in men. The index variants in these loci are further associated with related traits, including diabetes, hypertension, and cardiovascular disease. Lastly, we demonstrate that genetic risk scores constructed from significantly associated eGFR variants are correlated with CKD on a population level.

## Results

**Meta-analysis of eGFR.** Meta-analysis of 350,504 individuals (26,237,160 variants) from the HUNT Study, CKDGen Consortium, BioBank Japan, and the Michigan Genomics Initiative identified 147 loci associated with eGFR, of which 53 were novel (Table 1, Supplementary Data 1, Supplementary Fig. 1). We prioritized genes belonging to several biological classes related to kidney function based on: missense variants (either the lead or a proxy variant, 34 genes), DEPICT gene prioritization results (156 genes), significantly colocalized eQTLs in either kidney (4 genes) or non-kidney (187 genes) tissue, or nearby Mendelian kidney-disease genes (Supplementary Tables 1–2, Supplementary Data 2–4). We were able to prioritize genes using these annotations for 126 of the 147 loci (86%). Loci that were not able to be prioritized through these methods were annotated as the nearest gene (21/147 loci, 14%). Prioritized genes at novel loci included genes involved in transcription (*CASZ1, PPARGC1A, ZNF641, MED4-AS1, ZFHX3, ZGPAT, MAFF*), cellular signaling and differentiation (*ACVR2B, DCDC2, GRB10, THADA, TRIB1, PTPN3*), metabolism (*L2HGDH, XYLB*), solute carrier genes (*SLC25A43, TPCN2, KCNMA1, MFSD6*), and genes related to AB antigen

blood types (*ABO, FUT2*). Together, these results explain 7.6% of eGFR heritability, as calculated from LD score regression[21]. We were not able to directly test these variants for replication of the eGFR associations since a similarly-sized cohort with eGFR measurements was not available. Instead, we tested for association of the index variants in kidney-related traits in the UK Biobank (CKD, hypertensive CKD, renal failure, acute renal failure, renal failure NOS, renal dialysis, or other disorders of kidney and ureters). Seven of the 48 novel variants (5 were lost due to poor imputation) and 27 of the 85 lead variants in known loci that were available in the UK Biobank were at least nominally associated and had corresponding direction of effect with one or more UK Biobank kidney-related phenotypes, providing initial support for the biological validity of the eGFR results (Supplementary Data 5, Supplementary Fig. 2). In addition, we compared the results from the current meta-analysis with previously reported eGFR index variants[8–11,13,14,16]. Excluding the previously published datasets, 56 of the 118 available variants were at least nominally significant in a meta-analysis of HUNT and MGI alone (Supplementary Data 6).

**Kidney-specific eQTL associations.** To identify variants that may be acting through regulation of gene expression within the kidneys, we examined which eGFR index variants were significant eQTLs ($p$-value $< 6.7 \times 10^{-6}$, Bonferroni correction for 51 tissue types and 147 index variants) for a given gene in human kidney cortex[22], glomerulus[23], or tubulointerstitium. This identified 16 genes whose expression was associated with the eGFR index variants in kidney tissues, including 7 genes identified from normal kidney cortex tissue samples[22] and 10 genes identified from kidney glomerulus or tubulointerstitium samples[23] from individuals with nephrotic syndrome (1 gene overlapped both datasets, Supplementary Data 7). Five of these genes had expression levels associated with the eGFR index variants specifically in kidney tissues ($p$-value $< 6.7 \times 10^{-6}$) but not in other tissues in GTEx[24]: *APOD, CDKL5, DPEP1, FGF5,* and *TFDP2*. Of the kidney eQTLs, *FGF5, CDKL5, TPSAN33,* and *METTL10* showed significant colocalization with the eGFR association (Supplementary Data 3). In addition, some genes demonstrated a colocalizing eQTL association in non-kidney tissues but are also thought to cause Mendelian kidney diseases[25] when disrupted (*ALMS1, DCDC2, MUC1, RPS10, SDCCAG8, SLC34A1*).

**DEPICT analysis.** DEPICT analysis was performed to identify tissues and gene sets enriched for genes in the loci identified from eGFR meta-analysis. Consistent with the role of the identified genes in kidney function, the most significant tissues (FDR < 0.01) identified by DEPICT were the urinary tract and kidney (Supplementary Data 8). Additional enriched tissues (FDR < 0.05) included the exocrine glands, liver, epithelial cells, prostate, kidney cortex, male genitalia, membranes, and adrenal cortex. DEPICT analysis identified 482 significant gene sets (FDR < 0.05, Supplementary Data 9). The top gene sets ($p$-value $< 3.46 \times 10^{-6}$, 0.05/14462 gene sets) primarily included those associated with kidney morphology, the activity of transport channels, and with monosaccharide metabolic processes as shown in Fig. 1 (Supplementary Fig. 3).

**Overlap of eGFR loci with related traits.** As individuals with CKD often have coexisting heart disease or diabetes, we examined the identified eGFR variants for evidence of pleiotropic effects. A PheWAS analysis of the eGFR index variants across 23 cardiovascular and diabetes-related phenotypes in UK Biobank, excluding individuals with CKD, identified 7 phenotypes for which a subset of the index variants was also significant ($p$-value $< 1.48 \times 10^{-5}$,

**Table 1 Lead variants for novel eGFR loci from meta-analysis**

| Chr | Pos (hg19) | rsID | Ref | Alt | Freq[a] | N | P-value | Direction[a] | Prioritized genes |
|---|---|---|---|---|---|---|---|---|---|
| 1 | 10733081 | rs284316 | T | C | 0.3331 | 196273 | $1.50 \times 10^{-9}$ | + | CASZ1 |
| 1 | 100808363 | rs11166440 | A | G | 0.4119 | 350504 | $7.07 \times 10^{-9}$ | − | CDC14A |
| 1 | 180905694 | rs3795503 | T | C | 0.6017 | 350504 | $7.54 \times 10^{-13}$ | − | KIAA1614 |
| 1 | 227085824 | rs1800674 | A | G | 0.5615 | 350504 | $4.23 \times 10^{-8}$ | − | ADCK3 |
| 2 | 18679586 | rs10856778 | C | G | 0.8311 | 350504 | $1.04 \times 10^{-9}$ | + | LOC105373454 |
| 2 | 43441169 | rs35136921 | T | C | 0.4564 | 177995 | $2.74 \times 10^{-11}$ | + | THADA |
| 2 | 54574942 | rs1405833 | C | G | 0.2733 | 350504 | $5.10 \times 10^{-10}$ | − | C2orf73 |
| 2 | 178146362 | rs17581525 | C | G | 0.1889 | 350504 | $6.11 \times 10^{-11}$ | + | AC074286.1 |
| 2 | 191278341 | rs6725814 | A | G | 0.261 | 350504 | $3.39 \times 10^{-8}$ | + | MFSD6 |
| 2 | 230612451 | rs6756038 | A | G | 0.7353 | 350504 | $4.72 \times 10^{-9}$ | − | TRIP12 |
| 3 | 38479475 | rs7429308 | T | C | 0.4951 | 350504 | $4.90 \times 10^{-11}$ | − | ACVR2B, XYLB |
| 3 | 193816778 | rs10933714 | A | T | 0.5232 | 350504 | $2.81 \times 10^{-9}$ | − | LINC02028 |
| 3 | 195477791 | rs2291652 | A | G | 0.4385 | 347321 | $2.36 \times 10^{-8}$ | − | SDHAP2, MUC20, RP11-141C7.4, SDHAP1, MUC4, MIR570 |
| 4 | 23758662 | rs73243607 | T | C | 0.962 | 349244 | $3.37 \times 10^{-8}$ | − | PPARGC1A |
| 5 | 107459529 | rs12652687 | T | C | 0.1447 | 349666 | $3.61 \times 10^{-8}$ | − | FBXL17 |
| 6 | 24354045 | rs3765502 | T | C | 0.2351 | 347321 | $4.01 \times 10^{-8}$ | − | DCDC2 |
| 6 | 107172979 | rs7766720 | T | C | 0.1424 | 350503 | $4.13 \times 10^{-8}$ | − | LINC02532 |
| 7 | 50737852 | rs73116822 | T | C | 0.9282 | 349244 | $2.71 \times 10^{-9}$ | + | GRB10 |
| 7 | 56072841 | rs4948100 | T | C | 0.3796 | 350504 | $1.17 \times 10^{-8}$ | + | ZNF713, PSPH, CCT6A, GBAS |
| 7 | 128737958 | rs56088330 | A | T | 0.6814 | 350504 | $8.12 \times 10^{-10}$ | + | RP11-286H14.4, TSPAN33 |
| 8 | 9074223 | rs7006504 | T | C | 0.297 | 206846 | $1.91 \times 10^{-9}$ | − | PPP1R3B, ENSG00000254235, ENSG00000182319, RP11-10A14.5 |
| 8 | 32399662 | rs4489283 | T | C | 0.6897 | 349668 | $2.47 \times 10^{-9}$ | + | RP11-1002K11.1 |
| 8 | 126477978 | rs2001945 | C | G | 0.4361 | 350504 | $4.37 \times 10^{-11}$ | + | TRIB1 |
| 8 | 134332960 | rs10283362 | T | C | 0.8607 | 350504 | $2.04 \times 10^{-8}$ | − | NDRG1 |
| 9 | 34130435 | rs61237993 | A | G | 0.654 | 350501 | $1.85 \times 10^{-8}$ | − | DCAF12, UBE2R2 |
| 9 | 112206404 | rs10816812 | A | T | 0.6295 | 350501 | $1.00 \times 10^{-8}$ | + | PTPN3 |
| 9 | 136146597 | rs550057 | T | C | 0.7393 | 350501 | $1.58 \times 10^{-9}$ | − | ABO |
| 9 | 139107879 | rs11103387 | T | C | 0.7296 | 339928 | $8.07 \times 10^{-10}$ | − | QSOX2 |
| 10 | 35171118 | rs11010013 | A | G | 0.6928 | 323766 | $1.50 \times 10^{-8}$ | − | PARD3 |
| 10 | 79253261 | rs3127447 | A | C | 0.3283 | 349675 | $3.04 \times 10^{-8}$ | + | KCNMA1 |
| 10 | 94810665 | rs856534 | A | G | 0.4342 | 347320 | $1.23 \times 10^{-9}$ | + | EXOC6, CYP26C1 |
| 10 | 126418782 | rs11245344 | T | C | 0.4673 | 350504 | $3.37 \times 10^{-11}$ | + | METTL10 |
| 11 | 68883556 | rs7131509 | T | C | 0.5311 | 350503 | $2.76 \times 10^{-10}$ | − | TPCN2, LOC107984345 |
| 12 | 48736985 | rs2732481 | T | G | 0.2444 | 203655 | $1.82 \times 10^{-9}$ | − | ZNF641 |
| 13 | 48654455 | rs9534949 | C | G | 0.6165 | 350503 | $4.84 \times 10^{-9}$ | + | NUDT15, MED4-AS1, LINC00562 |
| 14 | 50735947 | rs72683923 | T | C | 0.0109 | 205585 | $2.98 \times 10^{-8}$ | + | SOS2, L2HGDH |
| 15 | 39274261 | rs8026431 | A | C | 0.6364 | 350504 | $3.13 \times 10^{-9}$ | − | LOC105370781 |
| 15 | 57830151 | rs117047297 | T | C | 0.9836 | 205586 | $3.16 \times 10^{-8}$ | − | CGNL1 |
| 15 | 63634405 | rs1075456 | T | G | 0.6601 | 350504 | $3.05 \times 10^{-9}$ | − | LACTB |
| 15 | 67561355 | rs12443279 | C | G | 0.3495 | 350504 | $3.03 \times 10^{-8}$ | − | SMAD3 |
| 16 | 69718112 | rs77944668 | A | G | 0.6984 | 350504 | $1.45 \times 10^{-8}$ | − | NFAT5, NQO1 |
| 16 | 73024276 | rs1858800 | T | C | 0.761 | 347323 | $1.82 \times 10^{-8}$ | − | ZFHX3 |
| 16 | 79938996 | rs35286975 | C | G | 0.7933 | 350504 | $1.94 \times 10^{-10}$ | − | MAF |
| 17 | 34950239 | rs12937411 | T | C | 0.6134 | 350502 | $1.62 \times 10^{-10}$ | − | MYO19, DHRS11 |
| 19 | 18384950 | rs1075403 | T | G | 0.6992 | 348391 | $1.57 \times 10^{-10}$ | − | JUND, KIAA1683 |
| 19 | 49217305 | rs281381 | A | G | 0.591 | 350504 | $4.77 \times 10^{-8}$ | + | RASIP1, FUT2 |
| 20 | 62336334 | rs1758206 | T | C | 0.8245 | 204319 | $1.51 \times 10^{-9}$ | + | ZGPAT, LIME1 |
| 21 | 16582710 | rs56038390 | A | G | 0.3014 | 350504 | $1.28 \times 10^{-9}$ | − | NRIP1 |
| 21 | 35356706 | rs2834317 | A | G | 0.8977 | 350504 | $1.58 \times 10^{-8}$ | + | LOC105372790 |
| 22 | 38600542 | rs2267373 | T | C | 0.3949 | 350504 | $2.65 \times 10^{-10}$ | + | MAFF |
| X | 18597869 | rs4825261 | A | C | 0.7148 | 96329 | $1.36 \times 10^{-9}$ | − | CDKL5 |
| X | 118630622 | rs454741 | A | G | 0.641 | 239987 | $5.15 \times 10^{-10}$ | − | SLC25A43, CXorf56 |
| X | 133808916 | rs11796053 | A | G | 0.4934 | 239987 | $4.93 \times 10^{-8}$ | + | HPRT1 |

[a]Reported frequency and direction of effect is with respect to the alternate allele for the combined meta-analysis
Study specific associations and allele frequencies are given in Supplementary Data 1
Gene names are italicized

Bonferroni correction for 23 phenotypes and 147 index variants): diabetes, coronary atherosclerosis, hypertension, essential hypertension, pulmonary heart disease, phlebitis and thrombophlebitis, and ischemic heart disease (Supplementary Data 5, 10, Fig. 2). Colocalization analysis with these phenotypes identified 7 loci (prioritized genes: FGF5, PRKAG2, TRIB1, DCDC5/MPPED2, L2HGDH/SOS2, UMOD, SALL1) having significantly colocalized association signals with hypertension, essential hypertension, and/or coronary atherosclerosis and 1 locus (prioritized gene: GCKR) that colocalized with association of type 2 diabetes (Supplementary

Data 5). Six of the seven index variants within loci that showed significant colocalization with the cardiovascular traits were associated with essential hypertension and/or hypertension, underscoring the connection between high blood pressure and CKD. In addition, the index variants were examined for association with 1,400 traits phenome-wide (without exclusion of CKD cases). As shown in Fig. 2, the index variants are significantly associated ($p$-value $< 5 \times 10^{-8}$) with additional traits including hypothyroidism and lipid metabolism disorders.

**Construction of genetic risk scores.** We developed genetic risk scores (GRS) from the meta-analysis results to assess the relationship between the identified variants and the likelihood of having CKD. These scores were then tested as predictors of CKD in a white British subset of UK Biobank. Several $p$-value and $r^2$ clumping thresholds were tested to select variants for inclusion in the GRS, but all yielded relatively similar predictions of CKD status (AUC range: 0.500–0.543). The best prediction was obtained using all independent markers ($r^2 < 0.4$) with $p$-value $< 5 \times 10^{-6}$ and the European subset of 1000 Genomes for LD clumping (Supplementary Fig. 4). The 1189 variants included in this risk score explain an estimated 25.3% of the variance in eGFR levels, while the GRS constructed using only the significantly associated independent variants ($p$-value $< 5 \times 10^{-8}$, $r^2 < 0.2$) is estimated to explain 9.4% of the variance in eGFR. The GRS alone was associated with CKD ($p$-value $= 1.8 \times 10^{-15}$, logistic regression), but it did not improve prediction of CKD status compared with birth year and sex alone (AUC: 0.700), or birth year, sex, and CKD clinical risk factors (diabetes, hypertension, and hyperlipidemia) (AUC: 0.865). Of all the tested models, inclusion of the GRS in addition to birth year, sex, and clinical risk factors provided the best predictor of CKD (AUC: 0.868). We also tested prediction of CKD using the best-performing GRS from the overall meta-analysis (without birth year or additional risk factors) separately in men and women. The GRS was slightly more predictive in women (AUC: 0.552) than in men (AUC: 0.538), possibly due to differing lifestyle or hormonal factors[26] between the sexes influencing the development of CKD. In summary, these results show that the variants identified from association studies of eGFR are correlated with the presence of CKD on a population level (Supplementary Fig. 4). However, the results are not sufficient to identify individuals with CKD from those without. This is consistent with findings from prior studies examining GRS of eGFR[27].

**Sex-specific analysis in HUNT.** We aimed to determine whether any of the eGFR index variants showed sex-specific association as there are known differences in the prevalence of CKD between men and women[28]. Association tests of eGFR in HUNT stratified by sex identified one locus (Fig. 3, Supplementary Fig. 5, Supplementary

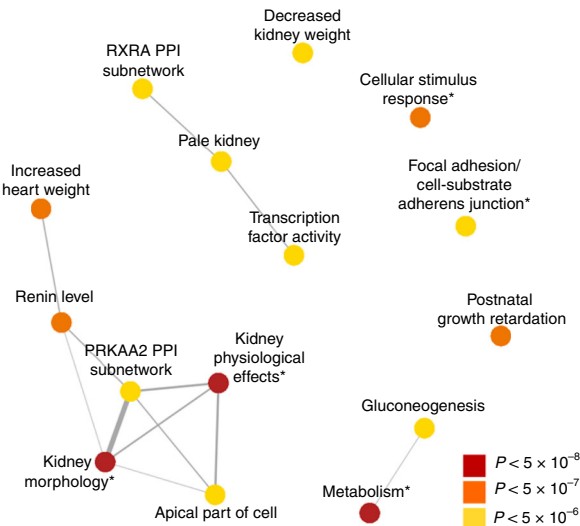

**Fig. 1** Top gene sets prioritized from eGFR meta-analysis. DEPICT analysis of eGFR meta-analysis results identifies significant gene sets associated with kidney function and metabolic processes. The most significant gene sets are shown ($p$-value $< 3.46 \times 10^{-6}$, 0.05/14462 gene sets, out of 482 with FDR $< 0.05$), after collapsing highly overlapping gene sets. Overlap between gene sets is depicted by the width of connecting lines. *Denotes collapsed gene sets

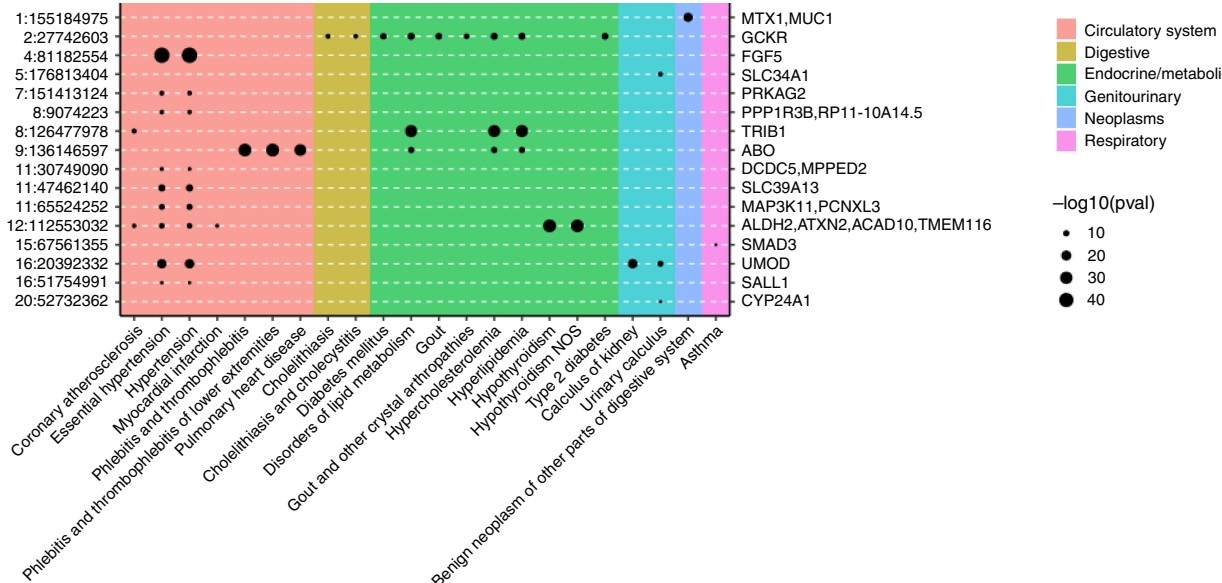

**Fig. 2** Pleiotropic associations of eGFR index variants. Index variants, given as chromosome:position on the left axis and prioritized gene on the right, from eGFR meta-analysis showing significant associations with at least one additional phenotype (16 variants with $p$-value $< 5 \times 10^{-8}$) in UK Biobank ($N_{max} =$ 408,961). 127 variants were tested for association with 1400 phenotypes

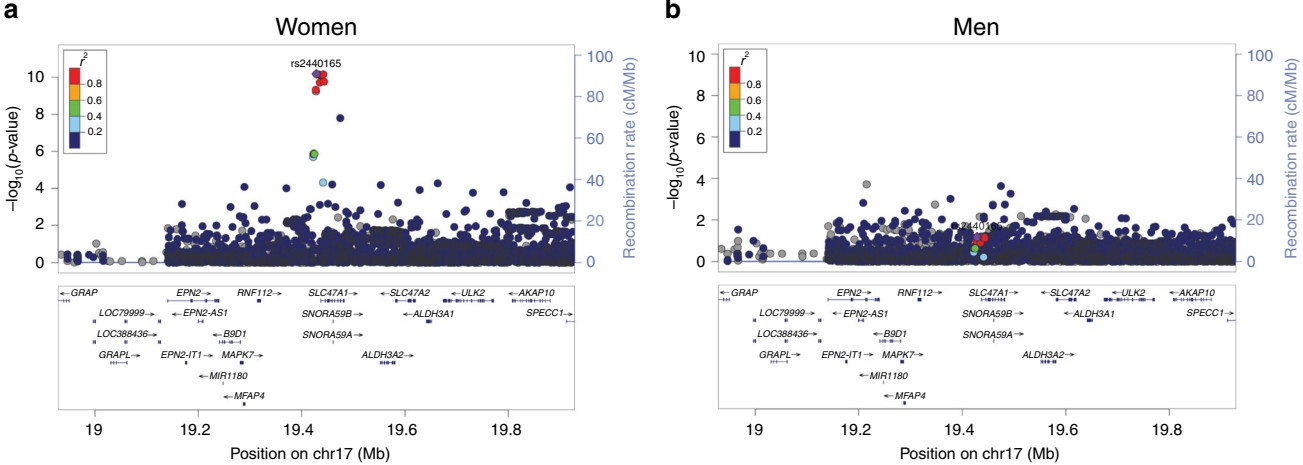

**Fig. 3** LocusZoom plots of region showing differential association between sexes. eGFR meta-analysis results in HUNT stratified by sex were filtered to identify regions significant in one sex ($P < 5 \times 10^{-8}$) but not significant ($P > 0.05$) in the other. Within each panel, we show regional results for eGFR association for variants near rs2440165, which is an eQTL for SLC47A1, in women (**a**) and men (**b**)

Data 11) that was significantly associated with eGFR in women but not in men. Interaction tests on the unrelated subset of individuals ($N = 26,235$, 37.7%) in HUNT confirmed a significant sex interaction for this variant ($p$-value $= 1.4 \times 10^{-5}$, $p$-value $< 0.05/147$). We obtained the summary statistics from the CKDGen consortium sex-stratified eGFR analysis to test this variant for replication[13]. Within the CKDGen results, the proxy variant of rs2440164, (rs2453580, $r^2 = 0.98$) showed greater significance and effect in women than in men ($p$-value$_{women} = 3.12 \times 10^{-5}$ effect$_{women} = -0.0066$, $p$-value$_{men} = 0.014$ effect$_{men} = -0.0055$).

## Discussion

In summary, analysis of the HUNT and MGI biobanks and meta-analysis of eGFR across more than 350,000 individuals identified 147 loci, of which 53 were novel. Novel eGFR index variants were common with relatively small effect sizes. Despite the small effect sizes of individual variants, the identified loci give new insight into the genes underlying kidney function and the development of CKD. In support of this, many of the prioritized genes cluster into known kidney associated pathways. For example, Wnt signaling has been implicated in kidney development and disease[29]. Variants in *DCDC2* were associated with eGFR levels. Knockdown or overexpression of DCDC2 is known to alter β-catenin activation of TCF transcription factors[30], thereby altering Wnt signaling. Likewise, variants in a gene associated with the epidermal growth factor receptor (ErbB) family were also observed. ErbB receptors are involved in kidney development[31], control of solute levels (e.g., Ca, Na)[32,33], and play a role in hypertension[34,35]. Our meta-analysis results identified variants in *MUC4* associated with eGFR levels. The beta chain of Mucin-4 (*MUC4*) interacts with ErbB2[36]. Lastly, we identified variants associated with both decreased eGFR and increased tubulointerstitial kidney CDKL5 expression. CDKL5 overexpression has been shown to impair ciliogenesis[37]. Defects in cilia are known to cause polycystic kidney disease and nephronophthisis, among other disorders[38]. In addition, prioritized genes included those known to cause Mendelian kidney disease[25]: *ALMS1, CNNM2, CYP24A1, CACNA1S, DACH1, DCDC2, GNAS, LRP2, MUC1, RPS10, SALL1, SCARB2, SDCCAG8, SHH, SLC34A1, SLC7A9, SMAD3,* and *UMOD*. These clues provide an initial link to how these identified genetic regions may lead to changes in kidney function.

Recent single-cell transcriptomic studies have classified kidney cell types in mice based on differential expression of specific genes

relative to other cell types[39]. Interestingly, several of the prioritized genes from the present study exhibited cell-type specific expression in mouse kidney: *CDC14A, DACH1,* and *VEGFA* in podocytes, *CGNL1, IRX1, PPP1R1B,* and *UMOD* in the loop of Henle, *LRP2, NAT8, SLC34A1, SLC47A1,* and *XYLB* in the proximal tubule, *RASIP1* in the endothelial, vascular, and descending loop of Henle, and *STC1* in collecting duct principal cells. These findings may help to identify kidney cell types whose function is affected by the genetic variants found in the eGFR GWAS. Recent studies have further examined the role of gene expression in eGFR and CKD. Xu et al., performed Mendelian randomization analysis using gene expression data to identify a causal role for *MUC1* expression on eGFR[40]. Single nucleus RNA-sequencing using cells from a human kidney donor identified expression of *DPEP1* that was specific to the proximal tubule[41]. Moreover, glomerular and tubular specific gene expression associations have been found to be significantly enriched for CKD and eGFR GWAS results[42], emphasizing the need to consider eQTLs in kidney tissue when prioritizing genes from kidney-related GWAS.

Experimental evidence also supports hormonal regulation of *SLC47A1* expression, the gene prioritized from the sex-stratified analysis of eGFR. *SLC47A1* is also known as *MATE1* (multidrug toxin and extrusion protein 1). Experimental studies of *MATE1* identified higher levels of expression in the kidneys of 30–45-day old male mice compared to female[43]. Furthermore, He at al. found that kidney expression of *MATE1* could be modified by treatment with testosterone or estradiol, compared to olive oil as a control[44].

It is also interesting to consider the interplay between kidney function and other related traits based on the overlap between identified genetic regions. For example, the *GCKR* gene was prioritized based on eGFR meta-analysis results. *GCKR* encodes glucokinase regulatory protein, which regulates glucose metabolism, and has been previously associated with the development of diabetes[45]. The *ABO* gene, responsible for determination of an individual's ABO blood type, was also prioritized based on eGFR meta-analysis results. Associations near this gene have been previously identified for other phenotypes, including LDL and total cholesterol[46], coronary artery disease[47], and type 2 diabetes[48]. As diabetes is a significant risk factor for the development of CKD, these shared associations may help to identify potential common mechanisms. Comparison of association results with cardiovascular disease-related traits also identified shared

associations with hypertension, the second major risk factor for CKD. Six of the 147 loci identified from meta-analysis showed significant colocalization with hypertension, which may help to identify additional shared pathways between high blood pressure and kidney function.

While additional studies are needed to understand eGFR associations that are specific to disease subtypes, the present results build upon the previous studies[8–18] to increase the number of eGFR associated loci and identify pleiotropic associations with cardiovascular disease. Limitations in the present study include the use of both population-based cohorts and cohorts selected for disease case status, differences in eGFR calculation and trait transformation between studies, and the lack of direct replication. Follow-up experimental studies are needed to validate the role of the identified genes in kidney function, and additional genetic studies are needed to verify these associations in more diverse cohorts. Nevertheless, these results identify additional genes that are likely involved in regulating kidney function and may help to identify new therapeutic targets or diagnostic measures to reduce the progression to CKD and need for permanent dialysis or kidney transplant.

## Methods

**Description of cohorts**. The HUNT study[49] is a longitudinal, repetitive population-based health survey conducted in the county of Nord-Trøndelag, Norway in which kidney-related phenotypes have not previously been tested for genetic association. Since 1984, the adult population in the county has been examined three times, through HUNT1 (1984–86), HUNT2 (1995–97), and HUNT3 (2006–08). A fourth survey, HUNT4 (2017–2019), is ongoing. HUNT was approved by the Data Inspectorate and the Regional Ethics Committee for Medical Research in Norway. All HUNT participants gave informed consent. Approximately 120,000 individuals have participated in HUNT1–HUNT3 with extensive phenotypic measurements and biological samples. A subset of these participants have been genotyped (~70,000) using Illumina HumanCoreExome v1.0 and 1.1 and imputed with Minimac3 using a combined HRC and HUNT-specific WGS reference panel. Variants with imputation $r^2 < 0.3$ were excluded from further analysis. We analyzed available kidney-related phenotypes within the HUNT study, including creatinine ($N = 69,591$), eGFR ($N = 69,591$), urea ($N = 20,700$), and CKD ($N_{cases} = 2044$ and $N_{controls} = 65,575$). eGFR values were calculated using the MDRD equation[50,51]. We also calculated eGFR using the CKD-EPI equation and after adjustment for covariates including age, sex, and batch followed by inverse normal transformation, the resultant eGFR phenotype values were highly correlated with those derived in the same manner after eGFR was calculated using MDRD (Supplementary Fig. 6, $r^2 = 0.995$). CKD status was derived from ICD-9 codes 585 and 586 and ICD-10 code N18. Association testing (24,961,484 variants) of quantitative traits was performed using BOLT-LMM[52] v2.2 on the inverse-normalized residuals of the traits adjusted for genotyping batch, sex, 4 principle components, and age (Supplementary Fig. 7). Association testing for CKD was performed using SAIGE with batch, sex, 4 principle components, and birth year as covariates. Associations stratified by sex for eGFR were also performed. For the stratified analyses, phenotypes were separately inverse-normalized and were adjusted for batch, age, and 4 principle components. Linkage disequilibrium within HUNT was calculated using PLINK v1.90[53]. To identify independent variants, conditional analysis for eGFR was performed within the HUNT dataset using BOLT-LMM v.2.3.1, conditioning on the lead variant within the identified loci until no variants with MAF > 0.5% had $p$-value $< 5 \times 10^{-8}$. To identify sex-specific effects, eGFR index variants were examined separately in men and women and were filtered to those that were significant in one sex but not significant ($p$-value > 0.05) in the other. Differences in effect sizes between males and females were tested using $Z = (\beta_M - \beta_W)/(SE_M{}^2 + SE_W{}^2 - 2rSE_M SE_W)^{0.5}$, where $r$ is the Pearson correlation for male and female effect sizes across all variants[54]. Interaction tests for these variants were performed on an unrelated subset of HUNT participants ($N = 26,235$) in PLINK v1.9 with age, sex, batch, and 4 principle components as covariates and a sex interaction term. Significance between sexes was determined using Bonferroni correction for the number of tested loci. HUNT-specific results are provided in Supplementary Tables 3–4 and in Supplementary Data 12.

BioBank Japan (BBJ) is a registry of patients from 12 medical centers across Japan who are diagnosed with at least one of 47 common diseases[55]. Summary statistics for 58 quantitative traits, including eGFR, are publicly available[16]. Participating individuals were genotyped with either the Illumina HumanOmniExpressExome BeadChip or HumanOmniExpress and HumanExome BeadChips. Imputation was performed with Minimac using the East Asian reference panel from 1000 Genomes phase 1[56]. Variants with imputation $r^2 < 0.7$

were excluded. BBJ eGFR values were calculated using the Japanese ancestry modified version of the CKD-EPI equation[57] and were available on 143,658 of those enrolled. Individuals with eGFR values of less than 15 mL/min/1.73 m$^2$ were excluded from the analysis. Values were standardized using rank-based inverse normalization. Association analysis (6,108,953 variants) was performed using mach2qtl with sex, age, the top 10 principle components, and disease status of all studied diseases ($N = 47$) included as covariates.

The CKDGen consortium includes meta-analysis results from 33 individual studies of European ancestry ($N = 110,527$) that were imputed with the 1000 Genomes phase I reference panel[8]. Summary statistics for eGFR were taken from the published dataset [ckdgen.imbi.uni-freiburg.de]. Detailed descriptions of individual cohorts are available[8]. Briefly, each group generated association statistics based on the natural log of eGFR using age and sex as covariates. eGFR was estimated from creatinine levels using the MDRD equation[50,51]. Variants with imputation quality ≤ 0.4, and those found in less than half of individuals were excluded from further analysis. Meta-analysis (10,154,908 variants) was performed using the inverse-variance method in METAL[58]. Pre and post-meta-analysis genomic control (GC) correction was performed.

The Michigan Genomics Initiative (MGI) is a repository of electronic medical record and genetic data at Michigan Medicine[59] ($N = 26,738$). MGI participants are enrolled during pre-surgical encounters at Michigan Medicine and consent to linkage of genetic and clinical data for research. MGI was approved by the Institutional Review Board of the University of Michigan Medical School. DNA was extracted from blood samples and then participants were genotyped using Illumina Infinium CoreExome-24 bead arrays. Genotype data was then imputed to the Haplotype Reference Consortium using the Michigan Imputation Server, providing 17 million imputed variants after standard quality control and filtering. Unrelated European individuals were used for analysis. eGFR values were computed using the CKD-EPI equation from creatinine values. The mean eGFR value was used for individuals having more than one eGFR measurement (median number of measurements per individual was 7 and the median time between first and last measurements was 2.4 years). 3% of individuals in MGI had a diagnosis of CKD. Due to the recruitment strategy of MGI, laboratory measurements are highly skewed towards more recent values, with 80% of laboratory values collected in 2010 or later. Mean eGFR was then regressed on sex, current age, array version, and PC1-4 and the subsequent residuals were inverse-normalized. Single-variant association testing of the inverse-normalized residuals was performed in *epacts* using a linear regression model for variants with MAF > 0.001 (12,560,917 variants).

**Meta-analysis**. Meta-analysis was performed using the $p$-value based approach in METAL[58]. This approach was chosen to account for differing units between the effect sizes of the CKDGen (log-transformed) and MGI/BBJ/HUNT (inverse-normalized) summary statistics. This approach was validated by comparison to traditional standard error-based meta-analysis of the cohorts with available inverse-normalized summary statistics; the results showed an extremely high correlation of $p$-values (Pearson $r = 0.966$, Supplementary Fig. 8). Summary statistics from contributing studies were GC corrected prior to meta-analysis and were not filtered by minor allele frequency or sample size. Lead index variants were determined as the most significant variant in ±1 Mb windows that were found in at least 2 studies. Adjacent windows were merged if the LD $r^2$ between lead variants was ≥ 0.2. Identified variants were considered to be novel if the most significant variant was more than 1 Mb away from previously reported lead variants. Linkage disequilibrium between variants was calculated using LDlink[60] or PLINK with the European and East Asian 1000 Genomes Phase III reference panels[61]. LD Score regression intercepts and heritability were calculated using LDSC version 1.0.0[21] with the European 1000 Genomes reference LD Scores using variants with minor allele frequency greater than 1%.

**Variant and gene annotation**. Variants were annotated using WGSA[62] and dbSNP[63]. Annotation of variants with associated biological processes was performed using the UniProt[64] and NCBI gene [https://www.ncbi.nlm.nih.gov/gene] databases. Genes for identified loci were prioritized based on the consensus between significantly colocalized eQTLs, missense variants within 1 Mb and in LD ($r^2 > \sim 0.8$) with the lead variant, and the gene prioritized by Data-driven Expression-Prioritized Integration for Complex Traits (DEPICT)[65]. In cases where there was no gene identified from the different annotation methods, the gene was prioritized as the nearest gene. DEPICT analysis was performed using the DEPICT 1.1 1000 Genomes version. Variants from meta-analysis that were found in two or more studies with $p$-value $< 5 \times 10^{-8}$ were included. LD information from the European and East Asian subsets of 1000 Genomes was used to construct loci within DEPICT. DEPICT results with FDR < 0.05 were considered significant. Gene sets with more than 25% overlap were collapsed into a single set for construction of the network diagram, as previously done[66].

**eQTL analysis**. Publicly available eQTL association datasets from GTEx V7[24], NephQTL[23], and Ko et al.[22] were each used separately to identify overlap between gene expression and identified eGFR association results. Specific tissue types,

sample sizes, and links to public datasets included in the analysis are given in Supplementary Table 2. Kidney eQTL results were taken from only NephQTL and the Ko et al. datasets due to the small sample size of the GTEx kidney dataset. NephQTL includes kidney samples from individuals with nephrotic syndrome and the Ko et al. dataset includes normal kidney samples from the Cancer Genome Atlas (TCGA). Lookup of individual variants for association with gene expression was performed using a Bonferroni-corrected $p$-value threshold of $6.7 \times 10^{-6}$ (correction for 51 tissue types and 147 index variants). The resulting associations were considered to be kidney-specific if an index variant was significantly associated with expression of a given gene in any of the kidney-specific datasets but not in other tissues available (from GTEx, Supplementary Fig. 9). Colocalization analysis was performed using the R package coloc[67]. Priors for p1, p2, and p12 within the coloc analysis were set to $1 \times 10^{-4}$, $1 \times 10^{-4}$, and $1 \times 10^{-6}$, respectively. Variants in the ±500 kb region surrounding each eGFR index variant were used for input into coloc. Within this region, we required at least one genome-wide significant ($p$-value $< 5 \times 10^{-8}$) eQTL variant prior to testing for colocalization. Following the criteria published by Giambartolomei et al.[67], eQTLs were considered to colocalize with the eGFR association results if the posterior probability (PP) for a shared variant was $> 80\%$.

**Determination of nearby Mendelian kidney disease genes.** Genes associated with Mendelian forms of kidney disease were taken from those identified by Groopman et al.[25] and from genes included in the KidneySeq v3 testing panel (Iowa Institute of Human Genetics). Nearby genes were identified as all genes having any overlap with the ± 250 kb region surrounding the index variants. Gene start and end positions were taken from the HAVANA gene annotations from the GENCODE consortium[68].

**Comparison with related traits.** Association results for phenome-wide lookups were taken from the publicly available analysis of UK Biobank[69] using SAIGE[70], which accounts for relatedness and population stratification by using a relationship matrix [http://pheweb.sph.umich.edu:5003/]. Phenotypes were grouped for analysis based on ICD-9 and 10 codes into phecodes following a similar strategy used by other groups for phenome-wide studies[71,72]. Additionally, we re-analyzed cardiovascular and diabetic traits in the white British subset of UK Biobank excluding CKD cases as has been previously suggested for identifying pleiotropic effects[59]. We performed analysis using individuals in the white British subset of UK Biobank that were included in the kinship calculation, excluding those identified as outliers based on the missingness rate and heterozygosity, and those missing from the UK Biobank phasing calculations. At the genotype level, we included all variants used for calculation of the kinship matrix and excluded variants after imputation with INFO score $< 0.3$ and variants not in the HRC imputation panel. Association testing was performed using SAIGE with sex, birth year, and 4 PCs as covariates. Sample sizes for all related traits are given in Supplementary Data 10. Colocalization analysis was performed using the R package coloc, with the priors for p1, p2, and p12 set to $1 \times 10^{-4}$, $1 \times 10^{-4}$, and $1 \times 10^{-6}$, respectively. Genetic regions for colocalization testing were defined as the most significant variant for eGFR in each locus ±500 kb. Within this region, we required at least one genome-wide significant ($p$-value $< 5 \times 10^{-8}$) variant for each trait prior to testing for colocalization. Variants were considered to colocalize if the probability for a common variant was greater than 80%.

**Genetic risk scores.** Variants were selected for inclusion in the genetic risk score (GRS) using the clumping procedure in PLINK with varying $r^2$ thresholds of 0.2, 0.4, 0.6, and 0.8 and $p$-value thresholds of $5 \times 10^{-8}$, $5 \times 10^{-6}$, $5 \times 10^{-4}$, and $5 \times 10^{-3}$ on the meta-analysis results from all cohorts. Because the meta-analysis included both European and East Asian samples, but the validation set included primarily European samples, we separately constructed the GRS using either the European only or European and East Asian subsets of 1000 Genomes Phase 3 for clumping in PLINK. Effect sizes were estimated from meta-analysis of the BioBank Japan, MGI, and HUNT results only due to the differing effect size units of the CKDGen consortium results. The proportion of variance explained by the GRS was calculated as the sum of $\beta^2(1-f)2f$ across all variants included in the risk score, where $\beta$ and $f$ are the effect size and frequency from meta-analysis of BioBank Japan, MGI, and HUNT. GRS were then calculated within UK Biobank as the sum of risk alleles carried by each individual weighted by the effect size of each variant. As decreased eGFR is predictive of increased CKD, the negative value of the resulting risk score was used for further analysis. GRS were then tested as predictors of CKD, either alone or as a logistic model including birth year, sex, and GRS or birth year, sex, GRS, diabetes, hypertension, and hyperlipidemia status (2625 cases and 396,923 controls). When fitting the logistic model for prediction of CKD, individuals in UK Biobank were randomly split into two halves, with one-half of individuals used for model fitting and the other half used for testing the model. Prediction ability was assessed by the area under the ROC curve (AUC).

**Data availability**

Data generated during analysis is available from the corresponding author upon reasonable request. Meta-analysis eGFR summary statistics are available here: http://csg.sph.umich.edu/willer/public/eGFR2018/.

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

## Acknowledgements

We thank all research participants in the HUNT study and in all included studies for their dedication towards improving human health. We thank Bethany Klunder for project management. We thank the CKDGen Consortium for publicly sharing CKD summary statistics and for making available sex-specific results for a subset of variants following our request. Funding for this study was provided by the National Institutes of Health to CJW (R01 HL127564, R35 HL135824). M.G.S. is supported by the Charles Woodson Clinical Research Fund, the Ravitz Foundation, and by National Institutes of Health RO1-DK108805. This research has been conducted using the UK Biobank Resource under application number 24460. The Nord-Trøndelag Health Study (The HUNT Study) is a collaboration between HUNT Research Center, Faculty of Medicine and Health Sciences, Norwegian University of Science and Technology (NTNU), Trøndelag County Council, Central Norway Regional Health Authority, and the Norwegian Institute of Public Health. The K.G. Jebsen center for genetic epidemiology is financed by Stiftelsen Kristian Gerhard Jebsen, Faculty of Medicine and Health Sciences Norwegian University of Science and Technology (NTNU) and the Liaison Committee for education, research and innovation in Central Norway. HUNT genotyping was financed by the National Institute of Health (NIH), the University of Michigan, The Norwegian Research Council, and Central Norway Regional Health Authority and the Faculty of Medicine and Health Sciences, Norwegian University of Science and Technology (NTNU). The authors acknowledge the University of Michigan Medical School Central Biorepository for providing biospecimen storage, management, and distribution services in support of the research reported in this publication.

## Author contributions

S.E.G., C.J.W, G.R.A. and K.H. designed the study. S.E.G., C.J.W, M.G.S., S.H., S.R., K.H. and W.E.H. drafted the manuscript. J.B.N., M.G.S., M.E.G., A.H.S., K.H., S.H., M.Z., L.G.F., S.K., D.F., D.L. and C.M.B characterized phenotypes of samples. M.G.S., S.E.G., J.B.N., S.R. and S.H. contributed to the discussion of biological mechanisms. S.E.G., J.B.N.,

M.Z., W.Z., L.G.F., I.S. and D.F. performed the statistical analysis of association data. G.R.A., S.L., H.M.K. and C.J.W. provided statistical oversight. J.B.N., M.E.G., A.H.S., W.E.H., S.K., C.M.B., K.H. and C.J.W. applied for and maintained regulatory (IRB) and other approvals. C.M.B., G.R.A. and K.H. are Principal Investigators of cohorts.

## Additional information

**Competing interests:** G.R.A. is an employee of Regeneron Pharmaceuticals. The remaining authors declare no competing interests.

