## [Peer Review File · Nature Communications]

Reviewer #1 (Remarks to the Author):

1. The sex differences in the incidence and progression of chronic kidney diseases have been examined in many experimental and clinical studies – the general consensus is that while sexual disparity exists, it is a male sex that shows greater susceptibility and faster progression of non-diabetic kidney disease than the opposite gender (please see for example J. Reckelhoff's and/or J. Neugarten's publications). The authors suggest the contrary in the introduction and their evidence is based on a reference to a book chapter. I think this requires a thorough re-examination.
2. The purpose and the details of prioritising genes within loci implicated in GWAS are not convincing. Initially, it appeared that this "functional prioritisation" was conducted to provide some input into DEPCIT. However, in Figure S7, the authors clearly demonstrate that DEPICT data was used actually as an input into the prioritisation analysis. It appears that the proximity of a gene to a sentinel variant was given the top priority in the "consensus" above the insights from DEPICT, eQTL studies and input from missense proxy SNP which is very surprising. There is a large body of evidence showing that the genes closest to the sentinel variant are frequently not the biological mediator of the identified associations. I think it would be sensible i) to report all the GWAS associations first using the gene closest to the sentinel SNP first (a common practice in GWAS), ii) conduct a separate "prioritisation" analysis to see how frequently the "prioritised" gene is different to the one implicated by proximity.
3. The UK Biobank is a very sensibly selected resource for the replication of the GWAS signals but without some additional information and clarifications it is very difficult to assess the value of this analysis. Firstly, the description of the methods pertaining to this important analysis is inadequate. All the details of the selection, quality control filters for both variants and the phenotypes should be reported in detail. As is, it is not clear if the authors have derived their phenotypic information only from self-reported data or took advantage from linked Hospital statistics and mortality data. In the absence of blood biochemistry data, this could increase the number of "cases" and thus improve the power of the analysis. It also appears to me that certain phenotypes have been unnecessarily analysed separately – :i.e. "renal failure" and "renal dialysis". I suggest that the authors take a full advantage of the data in UK Biobank but some thought should be given to the definition of "cases" and "controls".
4. The details of eQTL analysis are shown in insufficient details both in the results and methods section. Firstly, it should be first appreciated that the kidney is very poorly represented in GTEx – only about 30 samples exist with the RNA-seq-derived transcriptomic information and GWAS for the purpose of eQTL analysis. Were they included in this analysis? What were the other criteria of inclusion of a given tissue from GTEx in the analysis (normally only tissues with more than 100 samples are included). The kidney eQTL data come from two different sources – Ko's paper used Tissue Cancer Genome Atlas resource (largely normal kidney tissue secured from cancer nephrectomies) while the recently published Gillies' paper used kidney samples from patients with nephrotic syndrome. Did the authors examine each of these 2 kidney datasets separately and if yes, did the eQTL have to be demonstrated in both to satisfy the criteria of kidney-specific signal? The details of all eQTL analyses must be provided – what window was used, how the correction for multiple testing was calculated etc.
5. The analysis of colocalisation between CKD and cardiovascular/metabolic traits in UK Biobank is interesting. There is clear redundancy among the selected phenotypes (i.e. essential

hypertension and hypertension) and my comments above pertaining to UK Biobank-based analysis apply here as well.

6. The analysis of genetic risk scores clearly demonstrated that the variants associated with CKD in GWAS are effectively meaningless as predictive/diagnostic tools. It is only fair to fully acknowledge this message in the discussion in the context of other important insights that GWAS can provide.

7. The last section of the results section – analysis of Hunt seems a little de-attached from the rest.

8. Page 9 – HUNT Study is a population-based resource. I am not clear why eGFR values were calculated in the individuals recruited into HUNT using MDRD equation and not CKD-EPI equation. Have the authors utilised the insights from biochemical analysis of blood to define the CKD status of the HUNT individuals (or was it based exclusively on ICD codes)?

9. Table 1 – it would be helpful to see beta and SE together with P-values for each of 4 cohorts that contributed to the meta-analysis.

10. All tables require much more detailed legends.

Other comments:

11. That “eGFR levels below 60 mL/min/1.73m²...” (Page 4) characterise chronic kidney disease is not completely precise statement. Based on the currently used clinical definitions, CKD can be defined in patients with eGFR above this threshold in the presence of increased levels of ACR.

12. Diabetes and hypertension are not “chronic health conditions” (Page 4) but chronic diseases.

Reviewer #2 (Remarks to the Author):

The authors performed a meta-analysis on eGFR using two large published datasets of European ancestry individuals and of individuals from Japan, respectively, and included two additional smaller datasets of European ancestry.

Follow-up analyses included colocalization, lookup for different traits, and sex-stratified analyses. Although several new genetic associations with eGFR were revealed, a major drawback of this study is the lack of replication (except for the sex-stratified results). The alternatively conducted test for association of the index variants in kidney related traits in the UK Biobank addresses this problem only very limited because the tested endpoints differ from eGFR (including continuous traits vs. case/control status, renal failure vs. whole eGFR spectrum).

The authors state that genetic studies are needed to verify these associations in more diverse cohorts. Given that this study includes an almost equally sized sample set of EA individuals and individuals from Japan, it would be interesting to see the overlap and the differences in the eGFR associations between BBJ and the European ancestry studies which would be first step in this direction.

I have some additional comments and questions that should be addressed:

- Kidney-Specific eQTL Associations: As far as I understood, the significant eQTLs were constrained for not having eQTL in other tissue than kidney. The rationale for this approach is not quite clear. Variants that may have eQTLs across different tissues including kidney and would be excluded by this approach. Although this was a kidney specific eQTL analysis, it would be interesting to see also the results without removing loci that have eQTLs in non-kidney tissues.

- for the DEPICT results a Bonferroni correction was applied for both tissue and geneset enrichment, which is very conservative given that in Tables S8 and S9 a FDR is provided. Thus, the authors report only the tip of the iceberg. A commonly used FDR cutoff of 5% would be more informative while controlling the type I error. Pattaro et al. (2016) reported already tissue enrichment beyond the urogenital tract that at least partially overlap with the results of this study using the proposed FDR cutoff. The authors should be consistent regarding the naming of the reported results: according to Supplementary Table 8 they report a mix of tissue names and MeSH terms of different levels for the DEPICT results, i.e. both significant tissue enrichments belong to the urogenital system.

- prioritization in gene annotation: It seems not obvious that a missense variant in moderate LD ($r^2 > 0.3$) should be given a priority for gene selection. If the missense is really causal, I would expect a high LD with the lead variant, or the missense could be an independent variant which could be validated by a conditional analysis. I suggest the authors justify this criteria or increase the LD cut-off substantially. The prioritization by colocalization seems misleadingly named as eQTL in Suppl Figure 7. Finally, please provide the final selection criteria that was used to define a prioritized gene e.g. in Supplementary Table 1 and for the results of Table 2

- colocalization analyses of the lookup of traits in the UK Biobank: Which tissue was used for the colocalization analyses? Were the genome-wide or nominally significant associations in UK Biobank used as basis for the colocalization analysis? Did the colocalized genes per locus overlap with the eGFR colocalization results?

- The GRS lookup with CKD was performed in the UK Biobank sample. How was CKD status defined in this UK Biobank sample? How many CKD cases and controls were available? Please provide the ROC curves of the reported AUCs as Supplemental Figures (at least for the best performing GRS).

- lookup of significant hits in the UK Biobank for kidney-related phenotypes (Supplementary Table 5): Please provide the effect estimates in addition to the p-values. How many cases and controls for each UK Biobank trait were included in the analyses? Were the effect directions for the CKD associations in line with the ones from eGFR? Were the effect directions for hypertension as expected with respect to eGFR? Scatterplots of the effect sizes of the eGFR associations vs. selected UK Biobank traits would be informative.

- MGI study methods: Please provide information about the time range of multiple creatinine measurements per individual that were used to average the analyzed eGFR. Furthermore, were unrelated EA individuals used (indicated in the reference publication)? Please add this information to the methods.

- sex-specific associations in the HUNT study: please specify which of the genes were prioritized as functional candidates based on colocalization and which ones on missense on high LD. Please provide also the effects of the lookup in UK Biobank in addition to the reported p-values.

Furthermore, the statement in the Methods "loci were identified separately in men and women and were filtered to those that were significant in one sex but near nominal significance in the other" is imprecise: which p-value exactly was defined as "near nominal"?

- Novel loci revealed in HUNT: The two reported novel loci for CKD are not plausible based on the very low MAF in combination with the number of cases, and without replication performed. A suitable MAF or minor allele count filter should be applied to the meta-analysis results, esp. for CKD. It should be mentioned, that the identified rare stop-gain variant in the known locus PKD2 still needs to be replicated in independent studies.

- Please provide gene names in Supplementary Tables 6, 11, 13. In addition, in Supplementary Table 13 it would be of interest to see whether prioritized genes of the conditionally independent signals differ from the prioritized genes of the main signals

- Supplementary Table 1: please provide the in which order the datasets are listed in the "direction" column

- Discussion: The association of ABO with type 2 diabetes is rather weak compared to other phenotypes, therefore the postulated link of the ABO gene to CKD via type 2 diabetes is very speculative. Furthermore, given the design of this study (i.e. limiting the lookup in the UK Biobank to cardiovascular traits) and the small overlap of 6 out of 147 loci that colocalize with hypertension, I am not convinced that this study really "highlights" the pleiotropic associations with cardiovascular disease as stated. Therefore, this statement should be toned down.

- Where any known hits not significant in the current meta-analysis analysis?

- limitations of this study should be stated, such as mix of population-based cohorts and disease cohorts (i.e. BBJ); no replication; heterogeneity in phenotype generation (different eGFR formulas and trait transformations)

- In addition to the lambdas provided in the QQ plots, LD score regression intercepts should be reported for the meta-analyses to disentangle possible inflation from polygenecity.

- Suppl Fig 6: There is a branch of p-values that deviates from the median line, i.e. where the p-value based meta-analysis p-value is higher than in the standard-error approach. Can the authors explain this deviation, i.e. why only a subset of associations is affected by this?

- Suppl Fig 6: Please rephrase "overwhelmingly" in the figure description, as a high correlation is rather expected for these type of analyses

Reviewer #3 (Remarks to the Author):

The paper reports results from GWAS meta-analyses of eGFR that includes the HUNT study and MGI biobank, and two other GWAS/consortium data. They identified 147 loci including 53 novel loci. They also report sex-specific findings and associations of SNPs with multiple outcomes. There are several issues that need further clarifications. These include the strategy used for meta-analyses (minimum allele frequency, minimum number of samples contributing data), which can affect the number of identified associations, and the strategy for gene prioritization, which can affect downstream analyses. Sex-specific analyses should include interaction methods. Replication was performed using different phenotypes than those used for the discovery. Additional comments are shown below.

Introduction

2nd paragraph: CKD associations with lipids are inconsistent in studies and the statement needs to be supported by additional references.

3rd paragraph: the authors make an argument that HRC dense imputation enables discovery of new loci but a comparison of their results with imputed 1000 genomes reference panel is not included.

Results

Page 4, lines 97-98. The rationale for prioritization of genes for downstream analyses using the genes nearest to the index variant is not supported by functional evidence or the own authors' results shown in Table S4.

Page 5, lines 105-108. These results are not replication as the phenotypes are disease-related and not eGFR.

Lines 111-114, this replication is not valid given the overlap in samples with published studies.

Lines 121-122. Specify if the eQTL is from normal kidney tissue or individuals with CKD, and if human or mouse tissue.

Page 6, lines 146-147, more details on the samples used for colocalization analysis including ancestry are needed.

Lines 155-172. How much of the eGFR heritability and variance are explained by the SNPs? The results suggest very little overall effects.

Line 168 statement related to differing lifestyle or hormonal factors needs to be supported by references.

Sex-specific analyses. Results may be driven by small samples within strata. A better strategy is using interaction methods.

Page 7, lines 188-197, report sex-specific direction of effect for variants. Given different strategies for phenotype transformation across multiple studies, how this was compared to HUNT?.

Lines 199-208. Should you used a more stringent p-value threshold given multiple traits tested?

Methods

Page 9, lines 263-264. HUNT study description. Is the study a longitudinal cohort or a multiple cross-sectional study?. How many individuals were recruited and how many were genotyped? Which criteria were used for selection of people for genotyping.

Line 280, adjustments for batch suggest that there were multiple genotyping and potential different studies combined within the HUNT study. Please clarify.

Page 10, lines 314-325. For the MGI, more details are needed on the visits used for serum creatinine, number of measures and timeframe selected for eGFR measures. In addition, more details are needed on relatedness and how this was accounted in the analyses.

Lines 327-330. There were also differences in eGFR equation estimation.

Page 11, lines 334. The strategy for meta-analyses used results from at least 2 studies, so samples may be small for some of the findings, particularly for the variant with a MAF=0.01 (rs72683923). Please include the N for each variant in Table 1. Also highlight the prioritized strategy for genes for each SNP in this Table by including a column with the evidence.

What was the filter for allele frequency in analyses?

Line 343. The LD strategy for gene annotation and most significant variant within 1 Mb window are different, please clarify.

Line 355-258, include number of samples for each resource and if human or not. "Kidney association results" is a confusing term, please change to "eGFR associations".

Page 12, lines 375-376, the studies chosen have differences in eGFR estimating equation that can affect the betas.

Figure 2, the number of SNPs shown in the figure are less than 127 described in the legend, please clarify.

Table 1. Please include European and East Asian allele frequencies.

Supplementary figure 3. QQ plots, the sex-specific lambda results are the same to the 6th decimal, please revise.

Supplementary figures 4 and 5. Include lambdas.

Results from supplementary figure 6 need to be included in the discussion and described in methods. Are the reported results based on p-values or standard deviations?

Supplementary figure 7 is not needed.

Table S7. Include the source for eQTL. It looks like most are from GTEx tissues. Where are the results for the NephQTL and Ko et al.?

Table S10. This information should be included in methods instead of a table.

Reviewer #1 (Remarks to the Author):

1. The sex differences in the incidence and progression of chronic kidney diseases have been examined in many experimental and clinical studies – the general consensus is that while sexual disparity exists, it is a male sex that shows greater susceptibility and faster progression of non-diabetic kidney disease than the opposite gender (please see for example J. Reckelhoff's and/or J. Neugarten's publications). The authors suggest the contrary in the introduction and their evidence is based on a reference to a book chapter. I think this requires a thorough re-examination.

We have updated the introduction to clarify this point and have added additional references (Carrero et al, Nature Rev Neph 2018; Reckelhoff and Sampson, Am J Physiol Regul Integr Comp Physiol 2015; Neugarten J Am Soc Nephrol. 2002)

2. The purpose and the details of prioritising genes within loci implicated in GWAS are not convincing. Initially, it appeared that this “functional prioritisation” was conducted to provide some input into DEPCIT. However, in Figure S7, the authors clearly demonstrate that DEPICT data was used actually as an input into the prioritisation analysis. It appears that the proximity of a gene to a sentinel variant was given the top priority in the “consensus” above the insights from DEPICT, eQTL studies and input from missense proxy SNP which is very surprising. There is a large body of evidence showing that the genes closest to the sentinel variant are frequently not the biological mediator of the identified associations. I think it would be sensible i) to report all the GWAS associations first using the gene closest to the sentinel SNP first (a common practice in GWAS), ii) conduct a separate “prioritisation” analysis to see how frequently the “prioritised” gene is different to the one implicated by proximity.

We have updated the prioritization analysis to reflect the most likely gene as those identified by all three methods (DEPICT, eQTL colocalization, and missense $r^2 > 0.8$); if none, then genes identified by two methods; if none, then genes identified by any method; otherwise we list the nearest gene only when the listed approaches do not identify any candidate genes.

3. The UK Biobank is a very sensibly selected resource for the replication of the GWAS signals but without some additional information and clarifications it is very difficult to assess the value of this analysis. Firstly, the description of the methods pertaining to this important analysis is inadequate. All the details of the selection, quality control filters for both variants and the phenotypes should be reported in detail. As is, it is not clear if the authors have derived their phenotypic information only from self-reported data or took advantage from linked Hospital statistics and mortality data. In the absence of blood biochemistry data, this could increase the number of “cases” and thus improve the power of the analysis. It also appears to me that certain phenotypes have been unnecessarily analysed separately – :i.e. “renal failure” and “renal dialysis”. I suggest that the authors take a full advantage of the data in UK Biobank but some thought should be given to the definition of “cases” and “controls”.

Phecode analyses are an automated process by necessity of the number of ICD codes available in a complete EHR. Although we have spent months assessing and deriving phecodes in collaboration with the leaders in this area (Denny etc. at Vanderbilt), some fine-tuning may still be beneficial. We carefully re-reviewed the 26 phenotypes we focused on and removed 3 that had broad definitions and higher numbers of cases than expected.

We have identified cases of CKD in the UK biobank using ICD-9 codes 585 and 586 and ICD-10 code N18. We have aimed to be thorough in our lookup of related traits, and so some phenotypes are a subset of others, for example individuals on renal dialysis are a subset of individuals having renal failure. The scheme of ICD-9 and ICD-10 phecodes is now in a preprint server and we now cite this reference for detailed descriptions of the phecodes (<https://doi.org/10.1101/462077>).

4. The details of eQTL analysis are shown in insufficient details both in the results and methods section. Firstly, it should be first appreciated that the kidney is very poorly represented in GTEx – only about 30 samples exist with the RNA-seq-derived transcriptomic information and GWAS for the purpose of eQTL analysis. Were they included in this analysis? What were the other criteria of inclusion of a given tissue from GTEx in the analysis (normally only tissues with more than 100 samples are included). The kidney eQTL data come from two different sources – Ko's paper used Tissue Cancer Genome Atlas resource (largely normal kidney tissue secured from cancer nephrectomies) while the recently published Gillies' paper used kidney samples from patients with nephrotic syndrome. Did the authors examine each of these 2 kidney datasets separately and if yes, did the eQTL have to be demonstrated in both to satisfy the criteria of kidney-specific signal? The details of all eQTL analyses must be provided – what window was used, how the correction for multiple testing was calculated etc.

We are aware of the limitations of kidney eQTL data in GTEx and thus sought additional kidney eQTL datasets as noted by the reviewer. We have added additional information about the kidney eQTL analysis and criteria for significance in the methods. Also, we tested correlation between 111,739 SNP-eGene pairs common to our eQTLs and those that are publicly available from kidney cortex (unaffected parts of tumor nephrectomies). The Pearson correlation for GLOM-cortex and TI-cortex were 0.69 ($p < 2 \times 10^{-16}$) and 0.74 ($p < 2 \times 10^{-16}$), respectively. The vast majority showed consistent directional effect, albeit with effect-size heterogeneity.

5. The analysis of colocalisation between CKD and cardiovascular/metabolic traits in UK Biobank is interesting. There is clear redundancy among the selected phenotypes (i.e. essential hypertension and hypertension) and my comments above pertaining to UK Biobank-based analysis apply here as well.

Yes, however, the redundancy in phenotypes has a negligible impact on the multiple testing correction and, therefore, power. This is a standard PheWAS analysis.

6. The analysis of genetic risk scores clearly demonstrated that the variants associated with CKD in GWAS are effectively meaningless as predictive/diagnostic tools. It is only fair to fully acknowledge this message in the discussion in the context of other important insights that GWAS can provide.

It is unclear, at this time, if additional power from well-powered analyses of eGFR or CKD (using increased sample size and phenotype refinement as noted by the reviewer) may provide predictive utility of individuals at the tail of the risk distribution. Additional work is required in this area and we do not feel our study adequately provides evidence to claim there is no clinical utility of polygenic risk scores.

7. The last section of the results section – analysis of Hunt seems a little de-attached from the rest.

We have attempted to integrate these results better. Because HUNT is a new cohort where eGFR has not previously been analyzed (N=70k), we briefly describe the results in this cohort alone.

8. Page 9 – HUNT Study is a population-based resource. I am not clear why eGFR values were calculated in the individuals recruited into HUNT using MDRD equation and not CKD-EPI equation. Have the authors utilised the insights from biochemical analysis of blood to define the CKD status of the HUNT individuals (or was it based exclusively on ICD codes)?

CKD status was defined based on ICD codes.

9. Table 1 – it would be helpful to see beta and SE together with P-values for each of 4 cohorts that contributed to the meta-analysis.

We have added the study-specific results in supplementary table S1.

10. All tables require much more detailed legends.

These have been added.

Other comments:

11. That “eGFR levels below 60 mL/min/1.73m2...” (Page 4) characterise chronic kidney disease is not completely precise statement. Based on the currently used clinical definitions, CKD can be defined in patients with eGFR above this threshold in the presence of increased levels of ACR.

Thank you. We have corrected the text.

12. Diabetes and hypertension are not “chronic health conditions” (Page 4) but chronic diseases.

Thank you.

Reviewer #2 (Remarks to the Author):

The authors performed a meta-analysis on eGFR using two large published datasets of European ancestry individuals and of individuals from Japan, respectively, and included two additional smaller datasets of European ancestry.

Follow-up analyses included colocalization, lookup for different traits, and sex-stratified analyses. Although several new genetic associations with eGFR were revealed, a major drawback of this study is the lack of replication (except for the sex-stratified results). The alternatively conducted test for association of the index variants in kidney related traits in the UK Biobank addresses this problem only very limited because the tested endpoints differ from eGFR (including continuous traits vs. case/control status, renal failure vs. whole eGFR spectrum).

We agree that replication would be important to the field going forward and we have entered discussions with Andrew Morris, Christian Pattaro and Christian Fuchsberger to enter into a joint meta-analysis with these groups in the future. However, the work we present also highlights new results from 96,329 newly analyzed individuals and a new focus on sex-specific effects. We have also attempted to replicate the sex-specific eGFR findings in sex-specific CKD estimates.

The authors state that genetic studies are needed to verify these associations in more diverse cohorts. Given that this study includes an almost equally sized sample set of EA individuals and individuals from Japan, it would be interesting to see the overlap and the differences in the eGFR associations between BBJ and the European ancestry studies which would be first step in this direction.

Thank you for this suggestion and we have added a short discussion of the differences between the Japanese and European ancestry results.

I have some additional comments and questions that should be addressed:

- Kidney-Specific eQTL Associations: As far as I understood, the significant eQTLs were constrained for not having eQTL in other tissue than kidney. The rationale for this approach is not quite clear. Variants that may have eQTLs across different tissues including kidney and would be excluded by this approach. Although this was a kidney specific eQTL analysis, it would be interesting to see also the results without removing loci that have eQTLs in non-kidney tissues.

Thank you. We have now identified eQTLs that are present in kidney tissue, and not necessarily specific to kidney tissue.

- for the DEPICT results a Bonferroni correction was applied for both tissue and geneset enrichment, which is very conservative given that in Tables S8 and S9 a FDR is provided. Thus,

the authors report only the tip of the iceberg. A commonly used FDR cutoff of 5% would be more informative while controlling the type I error. Pattaro et al. (2016) reported already tissue enrichment beyond the urogenital tract that at least partially overlap with the results of this study using the proposed FDR cutoff. The authors should be consistent regarding the naming of the reported results: according to Supplementary Table 8 they report a mix of tissue names and MeSH terms of different levels for the DEPICT results, i.e. both significant tissue enrichments belong to the urogenital system.

We have now added more of the results by relaxing the threshold for inclusion to FDR 5%.

- prioritization in gene annotation: It seems not obvious that a missense variant in moderate LD ($r^2 > 0.3$) should be given a priority for gene selection. If the missense is really causal, I would expect a high LD with the lead variant, or the missense could be an independent variant which could be validated by a conditional analysis. I suggest the authors justify this criteria or increase the LD cut-off substantially. The prioritization by colocalization seems misleadingly named as eQTL in Suppl Figure 7. Finally, please provide the final selection criteria that was used to define a prioritized gene e.g. in Supplementary Table 1 and for the results of Table 2

Missense variants may not always be in high LD with the index variant if the frequencies are different (for example, R46L in PCSK9 has a frequency of 2.3% and is in r^2 of 0.03 with the initial lead variant at that locus:rs11206510 (Willer et al. 2008). However, we agree that most of the time, the 'causal' variant for an association signal will be either the lead SNP or in high LD with the lead SNP. We have removed Supplementary Figure 7, and have updated the Supplementary tables with the selection criteria for all prioritized genes. We now require the missense variant to be in $r^2 > 0.8$ with the lead variant to prioritize a gene.

- colocalization analyses of the lookup of traits in the UK Biobank: Which tissue was used for the colocalization analyses? Were the genome-wide or nominally significant associations in UK Biobank used as basis for the colocalization analysis? Did the colocalized genes per locus overlap with the eGFR colocalization results?

We only perform colocalization analysis for phenotype associations that reached our PheWAS significance threshold (requiring at least one variant in the ± 500 kb region around the eGFR index variant to have $p < 5 \times 10^{-8}$ for the tested UK Biobank trait). We are not sure how to interpret the reviewer's last sentence and wonder if the reviewer is asking whether the prioritized genes at each locus overlap with the gene identified by eQTL colocalization results? Since eQTL and DEPICT (which uses eQTL) was used to prioritize genes, the gene identification would not be independent.

- The GRS lookup with CKD was performed in the UK Biobank sample. How was CKD status defined in this UK Biobank sample? How many CKD cases and controls were available? Please provide the ROC curves of the reported AUCs as Supplemental Figures (at least for the best

performing GRS).

CKD status was defined using PheCodes. We have added methods for this to the manuscript. ROC curves are now supplemental figures as well as % with CKD for each PRS bin.

- lookup of significant hits in the UK Biobank for kidney-related phenotypes (Supplementary Table 5): Please provide the effect estimates in addition to the p-values. How many cases and controls for each UK Biobank trait were included in the analyses? Were the effect directions for the CKD associations in line with the ones from eGFR? Were the effect directions for hypertension as expected with respect to eGFR? Scatterplots of the effect sizes of the eGFR associations vs. selected UK Biobank traits would be informative.

We have added the number of cases and controls to Supplementary Table 10, and the direction for eGFR, CKD beta and se to Supplementary Table 5. We have added the scatter plots of eGFR effect size relative to hypertension effect and CKD effect as Supplementary Figure 3.

- MGI study methods: Please provide information about the time range of multiple creatinine measurements per individual that were used to average the analyzed eGFR. Furthermore, were unrelated EA individuals used (indicated in the reference publication)? Please add this information to the methods.

Added.

- sex-specific associations in the HUNT study: please specify which of the genes were prioritized as functional candidates based on colocalization and which ones on missense on high LD. Please provide also the effects of the lookup in UK Biobank in addition to the reported p-values.

We have clarified the source of evidence for biological candidate genes at each locus in new supplementary table 2.

Furthermore, the statement in the Methods "loci were identified separately in men and women and were filtered to those that were significant in one sex but near nominal significance in the other" is imprecise: which p-value exactly was defined as "near nominal"?

We have clarified this in the text ($P > 0.005$)

- Novel loci revealed in HUNT: The two reported novel loci for CKD are not plausible based on the very low MAF in combination with the number of cases, and without replication performed. A suitable MAF or minor allele count filter should be applied to the meta-analysis results, esp. for

CKD. It should be mentioned, that the identified rare stop-gain variant in the known locus PKD2 still needs to be replicated in independent studies.

We have added minor allele counts and frequency within cases and controls to Supplementary Table 14 and have updated the text to point out the need for replication of these variants.

- Please provide gene names in Supplementary Tables 6, 11, 13. In addition, in Supplementary Table 13 it would be of interest to see whether prioritized genes of the conditionally independent signals differ from the prioritized genes of the main signals

We have added this information to the tables.

- Supplementary Table 1: please provide the in which order the datasets are listed in the "direction" column

Added.

- Discussion: The association of ABO with type 2 diabetes is rather weak compared to other phenotypes, therefore the postulated link of the ABO gene to CKD via type 2 diabetes is very speculative. Furthermore, given the design of this study (i.e. limiting the lookup in the UK Biobank to cardiovascular traits) and the small overlap of 6 out of 147 loci that colocalize with hypertension, I am not convinced that this study really "highlights" the pleiotropic associations with cardiovascular disease as stated. Therefore, this statement should be toned down.

Thank you, we have made this revision.

- Where any known hits not significant in the current meta-analysis analysis?

Two (of 127) previously reported variants were not significant in the current meta-analysis. These are given in supplementary table 6.

- limitations of this study should be stated, such as mix of population-based cohorts and disease cohorts (i.e. BBJ); no replication; heterogeneity in phenotype generation (different eGFR formulas and trait transformations)

Thank you for this suggestion.

- In addition to the lambdas provided in the QQ plots, LD score regression intercepts should be reported for the meta-analyses to disentangle possible inflation from polygenecity.

These are provided.

- Suppl Fig 6: There is a branch of p-values that deviates from the median line, i.e. where the p-value based meta-analysis p-value is higher than in the standard-error approach. Can the authors explain this deviation, i.e. why only a subset of associations is affected by this?

This is a set of SNPs from a locus in which the frequency differs between Japanese and European individuals, thus the square root of the sample size is not as effective at estimating the standard error when the allele frequency differs.

- Suppl Fig 6: Please rephrase "overwhelmingly" in the figure description, as a high correlation is rather expected for these type of analyses

Thank you for the careful review.

Reviewer #3 (Remarks to the Author):

The paper reports results from GWAS meta-analyses of eGFR that includes the HUNT study and MGI biobank, and two other GWAS/consortium data. They identified 147 loci including 53 novel loci. They also report sex-specific findings and associations of SNPs with multiple outcomes. There are several issues that need further clarifications. These include the strategy used for meta-analyses (minimum allele frequency, minimum number of samples contributing data), which can affect the number of identified associations, and the strategy for gene prioritization, which can affect downstream analyses. Sex-specific analyses should include interaction methods. Replication was performed using different phenotypes than those used for the discovery. Additional comments are shown below.

Introduction

2nd paragraph: CKD associations with lipids are inconsistent in studies and the statement needs to be supported by additional references.

We have added additional references in support of this.

3rd paragraph: the authors make an argument that HRC dense imputation enables discovery of new loci but a comparison of their results with imputed 1000 genomes reference panel is not included.

We have instead provided a reference that HRC imputation covers more of the genome than 1000 genomes. We feel that the effort involved in using and comparing to an older and less dense imputation reference panel is not the best science.

Results

Page 4, lines 97-98. The rationale for prioritization of genes for downstream analyses using the genes nearest to the index variant is not supported by functional evidence or the own authors'

results shown in Table S4.

We only use the nearest gene if there is no other biological candidate available. Evidence suggests the nearest gene is the causal gene in ~50% of the loci.

Page 5, lines 105-108. These results are not replication as the phenotypes are disease-related and not eGFR.

We have modified the language to indicate we were examining whether variants involved in eGFR, a quantifiable measure of kidney function, are involved in chronic kidney disease. Although not a strict replication, this lends validity to the study and the eGFR association results reported.

Lines 111-114, this replication is not valid given the overlap in samples with published studies.

We have reported the association results for the current meta-analysis, which partially overlaps with previously reported studies, and for meta-analysis of HUNT and MGI alone which does not have overlap.

Lines 121-122. Specify if the eQTL is from normal kidney tissue or individuals with CKD, and if human or mouse tissue.

We have added some text to clarify these experimental details.

Page 6, lines 146-147, more details on the samples used for colocalization analysis including ancestry are needed.

We have added details to this section of the methods

Lines 155-172. How much of the eGFR heritability and variance are explained by the SNPs? The results suggest very little overall effects.

We have added an explanation of the heritability explained and how the loci can provide insight into biological underpinnings of disease, even if heritability or effect sizes are low. For example, variants in PCSK9 explain a small proportion of heritability of lipid phenotypes, but PCSK9-inhibitors have a dramatic effect on LDL cholesterol.

Line 168 statement related to differing lifestyle or hormonal factors needs to be supported by references.

We have added a reference for this.

Sex-specific analyses. Results may be driven by small samples within strata. A better strategy is using interaction methods.

We have also added this analysis. For the sex-specific loci, the variants were relatively common so the smallest stratum had between 6,895 and 7,911 individuals.

Page 7, lines 188-197, report sex-specific direction of effect for variants. Given different strategies for phenotype transformation across multiple studies, how this was compared to HUNT?.

P-value based meta-analysis was performed to look for consistently stronger associations in one sex than another. We have clarified this in the manuscript.

Lines 199-208. Should you used a more stringent p-value threshold given multiple traits tested?

We have used the standard genome-wide significance threshold of $p < 5 \times 10^{-8}$ for these analyses. We did not further adjust this threshold as each trait was analyzed separately.

Methods

Page 9, lines 263-264. HUNT study description. Is the study a longitudinal cohort or a multiple cross-sectional study?. How many individuals were recruited and how many were genotyped? Which criteria were used for selection of people for genotyping.

This is described in other publications and we have added the references. We have also added some details to the methods.

Line 280, adjustments for batch suggest that there were multiple genotyping and potential different studies combined within the HUNT study. Please clarify.

Because 70,000 samples from the HUNT study were genotyped, they were not genotyped in one enormous batch due to reagent batches etc.. Samples were randomized between batches and analyses were performed using batch as a covariate.

Page 10, lines 314-325. For the MGI, more details are needed on the visits used for serum creatinine, number of measures and timeframe selected for eGFR measures. In addition, more details are needed on relatedness and how this was accounted in the analyses.

We have added these details to the methods section.

Lines 327-330. There were also differences in eGFR equation estimation.

We have added a note of this difference to the discussion section.

Page 11, lines 334. The strategy for meta-analyses used results from at least 2 studies, so samples may be small for some of the findings, particularly for the variant with a MAF=0.01 (rs72683923). Please include the N for each variant in Table 1. Also highlight the prioritized strategy for genes for each SNP in this Table by including a column with the evidence. What was the filter for allele frequency in analyses?

We have edited supplementary table 1 to add this additional information and have clarified the methods section.

Line 343. The LD strategy for gene annotation and most significant variant within 1 Mb window are different, please clarify.

We used an LD r^2 cutoff of 0.2 for a pairwise comparison of the most significant index variants to ensure that regions with LD extending greater than 1 Mb were not mistakenly considered to be new loci. In annotating missense variants, we used an LD threshold of $r^2 > 0.8$ to identify all variants that may be in moderate LD with the index variant. We have included this in the methods section.

Line 355-258, include number of samples for each resource and if human or not. "Kidney association results" is a confusing term, please change to "eGFR associations".

Thank you, we have changed this term.

Page 12, lines 375-376, the studies chosen have differences in eGFR estimating equation that can affect the betas.

We have meta-analyzed only those studies which were inverse-normalized, so that betas are on a standardized scale.

Figure 2, the number of SNPs shown in the figure are less than 127 described in the legend, please clarify.

We have edited the figure legend to reflect that only variants with additional significant associations are plotted.

Table 1. Please include European and East Asian allele frequencies.

We have added these to supplementary table 1

Supplementary figure 3. QQ plots, the sex-specific lambda results are the same to the 6th decimal, please revise.

We have repeated this calculation and the values are correct as given. This is due to both analyses having the same median p-value.

Supplementary figures 4 and 5. Include lambdas.

We have added these

Results from supplementary figure 6 need to be included in the discussion and described in methods. Are the reported results based on p-values or standard deviations?

We have included a discussion of supplementary Figure 6 in the methods section, and have added a discussion of variants with differing p-values between the meta-analysis methods to the figure legend. The reported results are based on p-values.

Supplementary figure 7 is not needed.

We have removed this figure and have edited the supplementary tables to instead show the evidence for the gene prioritization.

Table S7. Include the source for eQTL. It looks like most are from GTEx tissues. Where are the results for the NephQTL and Ko et al.?

We have edited the supplementary tables to show the source of each tissue for the eQTL analysis.

Table S10. This information should be included in methods instead of a table.

As we have extended this table to include sample sizes and ICD-codes, we think it would be too lengthy to include in the methods section directly.

Reviewer #1 (Remarks to the Author):

- 1) The importance of gender as a risk factor for chronic kidney disease requires a more balanced interpretation. Not all studies reported so far showed the higher prevalence of chronic kidney disease in women than men. For example, data from Japan show clear male disadvantage in both age-related decline in kidney function and incidence of end-stage renal disease (Kidney International 2008;74:505-512 and Kidney International 1996;49:800-805).
- 2) Neither Table S2 nor the text on Page 5 specifies the tissue(s) in which the eQTL analyses were conducted. I suspect these are GTEX non-kidney tissue? If the kidney resources were used in this analysis how the discrepancy between the kidney and non-kidney tissue was interpreted?
- 3) UK Biobank analyses are in essence based on data generated by previously conducted genome-wide association studies conducted in UK Biobank. Did the authors apply any quality control filters at i) genotype, ii) participant, iii) phenotype-level prior to their analysis of association between eGFR variants and kidney/cardiovascular phenotypes?
- 4) How many variants of 53 novel CKD variants were available for look-up in UK Biobank? (On page 5 it reads "Thirteen of the 48 novel variants...")
- 5) There is lack of consistency in reporting the results of this analysis in Table S7. For "CKD chronic renal failure" beta, SE and P-values are reported, for other traits, just P-values. Why?
- 6) I do not think that the nominal significance level is right for the analyses in UK Biobank given the number of tests conducted. There must be some penalty for multiple testing. Had the authors triangulated individual-level data from multiple sources of information (self-reported, hospital statistics, mortality etc.) available in UK Biobank into one over-arching phenotype i.e. "history of chronic kidney disease", they would have had higher number of disease cases for "replication" purposes (and thus better power), "purer" controls, fewer phenotypes and less significant requirement of penalty for multiple testing.
- 7) Responses to my questions about the eQTL analysis are not sufficient (comment 4). The authors should answer every question asked in this section and the answers should also be incorporated in the manuscript.
- 8) The authors appear to have ignored my request to contemplate the meaning of their analysis on genetic risk scores in the discussion.

9) The authors have not responded explicitly to my question 8. MDRD equation is not the best choice to calculate eGFR in the population-based resource.

10) I am not clear why information derived from serum creatinine (that is clearly available) was not used in combination with ICD codes to define better the CKD status of HUNT individuals. In any case, I do not see much value in the last section of the results devoted exclusively to the HUNT study given that these subjects were already included in the meta-analysis of eGFR (first section of the manuscript), my reservations pertaining to CKD definition in this study and the level of novel discovery in particular in the absence of replication.

Reviewer #2 (Remarks to the Author):

In the revised version, most of my concerns were addressed or clarified, but some issues remain.

According to the conditional analyses methods and the Manhattan and QQ plots (Supplementary Figure 1), a $MAF > 0.5\%$ filter was applied for the eGFR GWAS in HUNT. Why was no filter applied for the CKD analyses in HUNT? The new CKD associations are not convincing given the very low MAF and MAC. Besides the required replication (which was also stated by the authors), it is likely that these results are rather statistical artefacts of the association model which would also explain the implausible large effect sizes.

The effect estimates for the association with CKD were added to Supplementary Table 5, but the effect estimates for the other traits are still missing. For hypertension, this information would be particularly helpful for interpreting the Supplementary Figure 2, i.e. which SNPs have a significantly positive effect on both increased eGFR and increased risk of hypertension. For all other traits listed in Supplementary Table 7, at least the effect direction should be provided (if the table becomes too complex by adding beta/SE) for better interpretation of both the pleiotropic effects and the colocalization results.

In addition, the description of Supplementary Figure 2 should include what the error bars represent (I assume the SE of the beta instead of e.g. a 95% CI).

The time range (in years) spanning the multiple creatinine measurements in the MGI study is still not provided, or were the measurements taken within a few days only? This information would be helpful to get an impression of possible declining effects of eGFR given that the mean value was used for the analysis.

Thanks for clarifying the trait colocalization with the UK Biobank traits and solving my misunderstanding of the analyses performed. I suggest to state in the paragraph "Overlap with Related Traits" more clearly that trait-based colocalization between eGFR and UK Biobank traits were performed, to avoid confusion with eQTL-based colocalization described in the preceding paragraph.

As reported in the revised version, the $\lambda=1.20$ of the HUNT eGFR GWAS is high (Suppl Figure 6). An LD score regression for the HUNT GWAS results should be provided, too.

Supplementary Tables 8 and 14: what does "snpeff ensemble summary" (column header) mean? What do the numbers in brackets in this column stand for?

Although effect alleles are provided in Supplementary Table 9 (eQTL), the corresponding effect directions are missing.

As far as I see, the question of Reviewer #1 regarding the calculation of eGFR in HUNT by the MDRD instead of the CKD-EPI formula was missed to answer.

Reviewer #3 (Remarks to the Author):

The authors have answered some but not all questions of this reviewer and additional clarifications are needed. The authors should restrain from making strong assertions that are beyond the scope of the findings, given their analyses are based on associations and predictive models.

Issues related to new text/data included in the revision:

1. A justification for a new co-author added to the paper is needed.
2. References added to justify sex-specific effects are 3 review articles, one of them related to cardiovascular disease. Original research related to the prevalence of CKD in women is preferable.
3. I am not sure we know if CKD progression is faster in men versus women. This is a strong statement and it should be supported by references from original articles in humans.
4. The sentence "Several other health conditions interact with kidney function." does not make sense.
5. Results: LD score regression estimation of heritability, which population was used for this estimation, European or Japanese?

6. Results: 13 novel variants had nominal replication in UK Biobank is NOT a strong supportive evidence for the biological validity of the eGFR findings. The results are based on p-values. At least a comparison on consistency in the direction of the association is needed.
7. Genetic risk scores explain 25% of eGFR variance in which dataset?
8. Statement related to results from ROC that have a slightly better prediction of CKD in women DO NOT support use of genetic risk to identify individuals at risk.

Prior questions:

9. Question related to strategy used for meta-analyses (minimum allele frequency, minimum number of samples contributing data) was not answered.
10. The literature on lipids and CKD is so far not supportive of association in humans. The citations are two review articles and an original study relating lipids to cardiovascular disease. Not sure why the authors want to stress lipids since these are not related to main findings.
11. Clearly state in the main text the % of genes prioritize using the different strategies and % that was nearby gene.
12. The sentence "Of the 127 previously reported index variants for eGFR, 125 were at least nominally significant in the present meta-analysis". As mentioned before, your meta-analyses include prior studies used for discovery so these results should not be reported.
13. Question related to kidney eQTLs from normal versus patients with nephrotic syndrome was not addressed. In the paragraph reporting eQTLs (Supplementary Table 9), it is still not clear which genes were eQTL for kidney cortex, tubulointerstitium or other. The source of the kidney eQTL is also not listed in the main text or Supplementary Table 9. Given nephrotic patients have CKD, the evidence for eQTL based on that data is less supportive for eGFR findings not related to CKD. Include the source of the kidney tissue by the genes in the main text and in table S9 (new column – normal versus nephrotic tissue). Include the paragraph where the new changes were done in the main document in your answer.
14. Phewas analyses need to account for the multiple test performed to avoid spurious associations.
15. Sex-specific analysis strategy is unconventional. The authors first tested genome-wide differences between men and women using a $p < 0.005$ for significance, then tested gene-sex interaction just for the three variants identified in the sex-stratified analyses. Follow-up analyses in the UK Biobank reporting associations with chest pain: this is a soft outcome, and the authors should examine the association with cardiovascular disease events instead.
16. Question related to the ancestry used for colocalization analyses not answered.

Second set of reviews: Point-by-point response to reviewer comments. Our response is indicated in blue and by indentation.

Reviewer #1 (Remarks to the Author):

1) The importance of gender as a risk factor for chronic kidney disease requires a more balanced interpretation. Not all studies reported so far showed the higher prevalence of chronic kidney disease in women than men. For example, data from Japan show clear male disadvantage in both age-related decline in kidney function and incidence of end-stage renal disease (Kidney International 2008;74:505-512 and Kidney International 1996;49:800-805).

We have removed this portion of the introduction to instead focus on CKD overall.

2) Neither Table S2 nor the text on Page 5 specifies the tissue(s) in which the eQTL analyses were conducted. I suspect these are GTEX non-kidney tissue? If the kidney resources were used in this analysis how the discrepancy between the kidney and non-kidney tissue was interpreted?

As previously requested by reviewer 2, we have included eQTL results that are significant in non-kidney tissues as well as kidney tissues. The tissue and gene for the eQTL analysis are given in Table S5, as requested by reviewer 3. We have now also updated Table S2 to specify which eQTL results are from kidney tissue and the text on Page 5 to specify that both kidney and non-kidney tissue eQTL results were included.

3) UK Biobank analyses are in essence based on data generated by previously conducted genome-wide association studies conducted in UK Biobank. Did the authors apply any quality control filters at i) genotype, ii) participant, iii) phenotype-level prior to their analysis of association between eGFR variants and kidney/cardiovascular phenotypes?

Yes, we did. We have added more detail about the genotype, individual and phenotype-level exclusions in the supplemental methods. The new text reads:

“We performed analysis using individuals in the white British subset of UK Biobank that were included in the kinship calculation, excluding those identified as outliers based on the missingness rate and heterozygosity, and those missing from the UK Biobank phasing calculations. At the genotype level, we included all variants used for calculation of the kinship matrix and excluded variants after imputation with INFO score < 0.3 and variants not in the HRC imputation panel.”

4) How many variants of 53 novel CKD variants were available for look-up in UK Biobank? (On page 5 it reads “Thirteen of the 48 novel variants...”)

We apologize for the lack of clarity. We have now revised the sentence to be more clear. It now reads as “Seven of the 48 novel variants (5 were lost due to poor imputation)...”.

The number of variants has changed as we have now considered direction of effect in addition to p-value.

5) There is lack of consistency in reporting the results of this analysis in Table S7. For “CKD chronic renal failure” beta, SE and P-values are reported, for other traits, just P-values. Why?

We apologize for this omission. For all traits, p-values, beta, and se are now reported in Supplementary Table 7.

6) I do not think that the nominal significance level is right for the analyses in UK Biobank given the number of tests conducted. There must be some penalty for multiple testing. Had the authors triangulated individual-level data from multiple sources of information (self-reported, hospital statistics, mortality etc.) available in UK Biobank into one over-arching phenotype i.e. “history of chronic kidney disease”, they would have had higher number of disease cases for “replication” purposes (and thus better power), “purer” controls, fewer phenotypes and less significant requirement of penalty for multiple testing.

We clearly state in the manuscript which additional phenotypes reach the more stringent genome-wide significance threshold, and which reach only nominal significance. We feel important to provide this information in a supplemental table to delve deeper into patterns of phenotypic heterogeneity across all loci. Text from manuscript: “A PheWAS analysis of the eGFR-index variants across 23 cardiovascular and diabetes-related phenotypes in UK Biobank, excluding individuals with CKD, identified 7 phenotypes for which a subset of the index variants was also significant (p-value < 1.48×10^{-5} , Bonferroni correction for 23 phenotypes and 147 index variants): diabetes, coronary atherosclerosis, hypertension, essential hypertension, pulmonary heart disease, phlebitis and thrombophlebitis, and ischemic heart disease (**Supplementary Tables 7,12, Figure 2**). Colocalization analysis with these phenotypes identified 7 loci (prioritized genes: *FGF5*, *PRKAG2*, *TRIB1*, *DCDC5/MPPED2*, *L2HGDH/SOS2*, *UMOD*, *SALL1*) having significantly colocalized association signals with hypertension, essential hypertension, and/or coronary atherosclerosis and 1 locus (prioritized gene: *GCKR*) that colocalized with association of type 2 diabetes (**Supplementary Table 7**). Six of the seven index variants within loci that showed significant colocalization with the cardiovascular traits were associated with essential hypertension and/or hypertension, underscoring the connection between high blood pressure and CKD. In addition, the index variants were examined for association with 1,400 traits phenome-wide (without exclusion of CKD cases). As shown in **Figure 2**, the index variants are significantly associated (p-value < 5×10^{-8}) with additional traits including hypothyroidism and lipid metabolism disorders.”

All individuals with mortality data attributed to either CKD or renal failure were included in the original phenotypes. We examined the impact of inclusion of self-reported kidney disease (questions were history of: 1) renal failure, 2) renal failure requiring dialysis, 3) renal failure not requiring dialysis) and found that the overarching phenotype of “history of renal failure” resulted in 237 additional cases, a 3.4% increase relative to the previous definition of renal failure based on ICD codes (N=6,985). We felt that the nominal increase, as well as uncertainty of a self-reported phenotype in the absence of an ICD-code, did not warrant an entire new GWAS.

7) Responses to my questions about the eQTL analysis are not sufficient (comment 4). The authors should answer every question asked in this section and the answers should also be incorporated in the manuscript.

R1 original comment #4. The details of eQTL analysis are shown in insufficient details both in the results and methods section. Firstly, it should be first appreciated that the kidney is very poorly represented in GTEx – only about 30 samples exist with the RNA-seq-derived transcriptomic information and GWAS for the purpose of eQTL analysis. Were they included in this analysis? What were the other criteria of inclusion of a given tissue from GTEx in the analysis (normally only tissues with more than 100 samples are included). The kidney eQTL data come from two different sources – Ko’s paper used Tissue Cancer Genome Atlas resource (largely normal kidney tissue secured from cancer nephrectomies) while the recently published Gillies’ paper used kidney samples from patients with nephrotic syndrome. **Did the authors examine each of these 2 kidney datasets separately and if yes, did the eQTL have to be demonstrated in both to satisfy the criteria of kidney-specific signal? The details of all eQTL analyses must be provided – what window was used, how the correction for multiple testing was calculated etc.**

Original response: We are aware of the limitations of kidney eQTL data in GTEx and thus sought additional kidney eQTL datasets as noted by the reviewer. We have added additional information about the kidney eQTL analysis and criteria for significance in the methods. New text:

New response: We apologize for the lack of detail previously added. We considered kidney eQTL results to be those found in either the NephQTL and Ko et al. datasets and did not require significance in both. We have added additional text to state that these were used separately in the methods section. As also described in the methods section, we used a Bonferroni corrected significance threshold (6.7×10^{-6}) for testing of individual variants and a stricter threshold of 5×10^{-8} within a ± 500 kb window around the eGFR index variant when performing colocalization analysis. This paragraph now reads as:

“Publicly available eQTL association datasets from GTEx V7²⁴, NephQTL²³, and Ko et al.²² were each used separately to identify overlap between gene expression and identified eGFR association results. Specific tissue types, sample sizes, and links to public datasets included in the analysis are given in **Supplementary Table 4**. Kidney specific eQTL results were taken from only NephQTL and the Ko et al. datasets due to the small sample size of the GTEx kidney dataset. NephQTL includes kidney samples from individuals with nephrotic syndrome and the Ko et al. dataset includes normal kidney samples from the Cancer Genome Atlas (TCGA). Lookup of individual variants for association with gene expression was performed using a Bonferroni-corrected p-value threshold of 6.7×10^{-6} (correction for 51 tissue types and 147 index variants). The resulting associations were considered to be kidney-specific if an index variant was significantly-associated with expression of a given gene in any of the kidney-specific datasets but not in other tissues available (from GTEx). Colocalization analysis was performed using the R package coloc⁶⁴. Priors for p1, p2, and p12 within the coloc analysis were set to 1×10^{-4} , 1×10^{-4} , and 1×10^{-6} , respectively. Variants in the ± 500 kb region surrounding each eGFR

index variant were used for input into coloc. Within this region, we required at least one genome-wide significant (p -value $< 5 \times 10^{-8}$) eQTL variant prior to testing for colocalization. Following the criteria published by Giambartolomei et al.⁶⁴, eQTLs were considered to colocalize with the eGFR association results if the posterior probability (PP) for a shared variant was $> 80\%$. “

8) The authors appear to have ignored my request to contemplate the meaning of their analysis on genetic risk scores in the discussion.

The original request was to “acknowledge .. in the discussion” that “the variants associated with CKD in GWAS are effectively meaningless as predictive/diagnostic tools”. We respectfully disagree and explained why in our initial response and here again. AUC does not completely capture the predictive value of the PRS, particularly at the tails of the distribution. We have included a figure (Supplementary Figure 4) demonstrating the prevalence of CKD based on the PRS centile to provide evidence of our claim.

Supplementary Figure 4 legend excerpt: “CKD prevalence in the highest GRS percentile (1.05%) was 2.5 times higher than that of the lowest GRS percentile (0.43%) and 1.6 times higher than CKD prevalence among all other percentiles (0.65%).”

We believe that the results section sufficiently explains this, and the section relevant to this point currently reads: “In summary, these results show that the variants identified from association studies of eGFR are correlated with the presence of CKD on a population level and may be used to identify the individuals at highest risk (**Supplementary Figure 4**). However, the results are not sufficient to identify individuals with CKD from those without. This is consistent with findings from prior studies examining GRS of eGFR²⁷”

9) The authors have not responded explicitly to my question 8. MDRD equation is not the best choice to calculate eGFR in the population-based resource.

We apologize, this response appears to have been lost during edits of our response to reviewers last round. Because we performed inverse-normal transformation of residuals adjusted for covariates (age, sex, batch), we expected that using MDRD or CKD-EPI equation would have little impact on the phenotype used for association analysis. Indeed, we repeated the eGFR using CKD-EPI and found the correlation in the inverse-normal transformed residuals adjusted for covariates to be highly correlated ($r^2 = 0.995$). We have added the Figure below as Supplemental Figure 6 and added the following text to the methods “We also calculated eGFR using the CKD-EPI equation and after adjustment for covariates including age, sex, and batch followed by inverse normal transformation, the resultant eGFR phenotype values were highly correlated with those derived in the same manner after eGFR was calculated using MDRD (Supplementary Figure 6, $r^2 = 0.995$).”

10) I am not clear why information derived from serum creatinine (that is clearly available) was not used in combination with ICD codes to define better the CKD status of HUNT individuals. In any case, I do not see much value in the last section of the results devoted exclusively to the HUNT study given that these subjects were already included in the meta-analysis of eGFR (first section of the manuscript), my reservations pertaining to CKD definition in this study and the level of novel discovery in particular in the absence of replication.

We have removed this paragraph. Since we are authors from the HUNT study and this is our first publication on the HUNT study for kidney-related phenotypes, we had hoped to describe this large resource (N~70,000) but we defer to the reviewer and we now only describe results from the meta-analysis.

Reviewer #2 (Remarks to the Author):

In the revised version, most of my concerns were addressed or clarified, but some issues remain.

According to the conditional analyses methods and the Manhattan and QQ plots (Supplementary Figure 1), a MAF>0.5% filter was applied for the eGFR GWAS in HUNT. Why was no filter applied for the CKD analyses in HUNT? The new CKD associations are not convincing given the very low MAF and MAC. Besides the required replication (which was also stated by the authors), it is likely that these results are rather statistical artefacts of the association model which would also explain the implausible large effect sizes.

We suspect that the reason we identify lower frequency results in HUNT is because previous GWAS studies have only examined variants above a certain MAF threshold. Since the reviewer is skeptical of the findings with lower MAF and we cannot replicate these results in the short time frame, we have removed discussion of these results from the manuscript and include them only in the supplementary information.

The effect estimates for the association with CKD were added to Supplementary Table 5, but the effect estimates for the other traits are still missing. For hypertension, this information would be particularly helpful for interpreting the Supplementary Figure 2, i.e. which SNPs have a significantly positive effect on both increased eGFR and increased risk of hypertension. For all other traits listed in Supplementary Table 7, at least the effect direction should be provided (if the table becomes too complex by adding beta/SE) for better interpretation of both the pleiotropic effects and the colocalization results.

In addition, the description of Supplementary Figure 2 should include what the error bars represent (I assume the SE of the beta instead of e.g. a 95% CI).

We have added these effect estimates (beta, se, direction) for other phenotypes to Supplementary Table 5. We apologize for the oversight. We have added to Supplementary Figure 2 a description of the error bars: *"Error bars represent ± 1 SE."*

The time range (in years) spanning the multiple creatinine measurements in the MGI study is still not provided, or were the measurements taken within a few days only? This information would be helpful to get an impression of possible declining effects of eGFR given that the mean value was used for the analysis.

We have edited the text to report that the median time range between the first and last creatinine measurement was 2.4 years.

Thanks for clarifying the trait colocalization with the UK Biobank traits and solving my misunderstanding of the analyses performed. I suggest to state in the paragraph "Overlap with Related Traits" more clearly that trait-based colocalization between eGFR and UK Biobank traits were performed, to avoid confusion with eQTL-based colocalization described in the preceding paragraph.

Thank you for this suggestion. We have revised the heading.

As reported in the revised version, the $\lambda=1.20$ of the HUNT eGFR GWAS is high (Suppl Figure 6). An LD score regression for the HUNT GWAS results should be provided, too.

This is now provided. The LDSC intercept was 1.098.

Supplementary Tables 8 and 14: what does "snpeff ensemble summary" (column header) mean? What do the numbers in brackets in this column stand for?

Annotations were performed using WGS, and we report the SnpEff variant annotations. The numbers in brackets are the number of transcripts and number of affected transcripts for each gene and variant annotation, respectively. We have added descriptions of this to the Supplementary Tables. E.g. Supplementary Table 2 footnote: *"Annotation of variants was done using the SnpEff ensemble summary from WGS. The numbers in brackets denote the number of transcripts and number of affected transcripts for each gene and annotation type, respectively."*

Although effect alleles are provided in Supplementary Table 9 (eQTL), the corresponding effect directions are missing.

We have now provided the direction. We apologize for this oversight.

As far as I see, the question of Reviewer #1 regarding the calculation of eGFR in HUNT by the MDRD instead of the CKD-EPI formula was missed to answer.

Thank you. We have responded to reviewer 1 above.

Reviewer #3 (Remarks to the Author):

The authors have answered some but not all questions of this reviewer and additional clarifications are needed. The authors should restrain from making strong assertions that are beyond the scope of the findings, given their analyses are based on associations and predictive models.

Issues related to new text/data included in the revision:

1. A justification for a new co-author added to the paper is needed.

Because our first author is on maternity leave, an additional author was recruited to help address revisions.

2. References added to justify sex-specific effects are 3 review articles, one of them related to cardiovascular disease. Original research related to the prevalence of CKD in women is

preferable.

We have removed the discussion of differences in CKD prevalence between sexes from the introduction and instead focus on the prevalence of CKD overall.

3. I am not sure we know if CKD progression is faster in men versus women. This is a strong statement and it should be supported by references from original articles in humans.

We have removed this statement from the manuscript

4. The sentence "Several other health conditions interact with kidney function." does not make sense.

Thank you, we have revised this sentence. It now reads: "*Several other health conditions affect kidney function. Chronic diseases such as diabetes and hypertension directly influence the development of CKD, with environmental factors such as smoking accelerating disease progression⁵.*"

5. Results: LD score regression estimation of heritability, which population was used for this estimation, European or Japanese?

We have clarified this in the text to state that the European 1000 Genomes reference panel was used. We repeated the analysis using LD estimated from either 1) East Asian 1000G samples or 2) European 1000G and found similar results; 1) East Asian: 7.1% and 2) European: 7.6%.

6. Results: 13 novel variants had nominal replication in UK Biobank is NOT a strong supportive evidence for the biological validity of the eGFR findings. The results are based on p-values. At least a comparison on consistency in the direction of the association is needed.

We have edited the text to include only those results which were directionally consistent with kidney outcome (ie. Decreasing eGFR association and increasing kidney disease association) and at least nominally significant, and have removed the term "strong". The text now reads "*Seven of the 48 novel variants (5 were lost due to poor imputation) and 27 of the 85 lead variants in known loci that were available in the UK Biobank were at least nominally associated and had corresponding direction of effect with one or more UK Biobank kidney-related phenotypes, providing support for the biological validity of the eGFR results (Supplementary Table 7, Supplementary Figure 2).*"

7. Genetic risk scores explain 25% of eGFR variance in which dataset?

We have included in the methods section a statement that the variance explained by the risk scores was calculated using the effect sizes and frequencies from meta-analysis of BioBank Japan, MGI, and HUNT. The variance explained was estimated using $2pq \cdot \beta^2$ where p is the minor allele frequency, q is the major allele frequency and β is the per-allele effect estimate. The text reads: *“The proportion of variance explained by the GRS was calculated as the sum of $\beta^2(1-f)2f$ across all variants included in the risk score, where β and f are the effect size and frequency from meta-analysis of BioBank Japan, MGI, and HUNT. GRS were then calculated within UK Biobank as the sum of risk alleles carried by each individual weighted by the effect size of each variant.”*

8. Statement related to results from ROC that have a slightly better prediction of CKD in women DO NOT support use of genetic risk to identify individuals at risk.

We did not intend to say that the differences in prediction by sex support the use of GRS, but rather that GRS in general may be used to stratify individuals by risk level (Supplementary Figure 4). We have edited this sentence to clarify this point. The text reads:

“We also tested prediction of CKD using the best-performing GRS from the overall meta-analysis (without birth year or additional risk factors) separately in men and women. The GRS was slightly more predictive in women (AUC: 0.552) than in men (AUC: 0.538), possibly due to differing lifestyle or hormonal factors³⁴ between the sexes influencing the development of CKD. In summary, these results show that the variants identified from association studies of eGFR are correlated with the presence of CKD on a population level and may be used to identify the most at risk individuals (Supplementary Figure 4).”

Prior questions:

9. Question related to strategy used for meta-analyses (minimum allele frequency, minimum number of samples contributing data) was not answered.

We have included in the methods section a statement that we did not apply a pre- meta-analysis allele frequency or sample size filter. The relevant sentence is: *“Summary statistics from contributing studies were GC corrected prior to meta-analysis and were not filtered by minor allele frequency or sample size.”*

10. The literature on lipids and CKD is so far not supportive of association in humans. The citations are two review articles and an original study relating lipids to cardiovascular disease. Not sure why the authors want to stress lipids since these are not related to main findings.

We agree that the association of kidney disease with lipids is outside of the scope of this manuscript and have removed this sentence.

11. Clearly state in the main text the % of genes prioritize using the different strategies and % that was nearby gene.

We have added this to the results section: *“We were able to prioritize genes using these annotations for 126 of the 147 loci (86%). Loci that were not able to be prioritized through these methods were annotated as the nearest gene (21/147 loci, 14%).”*

12. The sentence “Of the 127 previously reported index variants for eGFR, 125 were at least nominally significant in the present meta-analysis”. As mentioned before, your meta-analyses include prior studies used for discovery so these results should not be reported.

We now only report results excluding previously published datasets: *“Excluding the previously published datasets, 56 of the 118 available variants were at least nominally significant in meta-analysis of HUNT and MGI alone (Supplementary Table 8).”*

13. Question related to kidney eQTLs from normal versus patients with nephrotic syndrome was not addressed. In the paragraph reporting eQTLs (Supplementary Table 9), it is still not clear which genes were eQTL for kidney cortex, tubulointerstitium or other. The source of the kidney eQTL is also not listed in the main text or Supplementary Table 9. Given nephrotic patients have CKD, the evidence for eQTL based on that data is less supportive for eGFR findings not related to CKD. Include the source of the kidney tissue by the genes in the main text and in table S9 (new column – normal versus nephrotic tissue). Include the paragraph where the new changes were done in the main document in your answer.

We had previously added Supplementary Table 4 which provides the samples sizes and data sources for all eQTL datasets (both kidney and non-kidney). This is noted at the top of Supplementary Table 9. We have now additionally added a column to Supplementary Table 9 noting the source of the eQTL data and for kidney tissues whether the tissue is from normal or nephrotic kidney samples. This paragraph in the main text has been edited to read: *“This identified 16 genes whose expression was associated with the eGFR index variants in kidney tissues, including 7 genes identified from normal kidney cortex tissue samples²² and 10 genes identified from kidney glomerulus or tubulointerstitium samples²³ from individuals with nephrotic syndrome (1 gene overlapped both datasets, Supplementary Table 9).”*

14. Phewas analyses need to account for the multiple test performed to avoid spurious associations.

We had previously used a genome-wide significance threshold of 5×10^{-8} when identifying pleiotropic associations, but have now edited the analysis to use a Bonferroni-corrected threshold of $p\text{-value} < 1.48 \times 10^{-5}$ (correction for 127 variants and 1400 traits).

15. Sex-specific analysis strategy is unconventional. The authors first tested genome-wide differences between men and women using a $p < 0.005$ for significance, then tested gene-sex interaction just for the three variants identified in the sex-stratified analyses. Follow-up analyses in the UK Biobank reporting associations with chest pain: this is a soft outcome, and the authors should examine the association with cardiovascular disease events instead.

We have changed the sex-specific analysis strategy to the reviewer’s preferred approach and have now analyzed all eGFR meta-analysis index variants using interaction tests. Due to the larger number of loci included and the change to eGFR meta-analysis index variants rather than those from HUNT, only one locus remains significant after Bonferroni correction for the 147 index variants. The association with chest pain is therefore removed from the text.

16. Question related to the ancestry used for colocalization analyses not answered.

Colocalization analysis was performed using the entire eGFR meta-analysis results (ie. 59% European and 41% East Asian individuals) with the white British subset of UK Biobank used for GWAS analysis of related traits. The PheWAS analysis was pre-calculated with white British individuals from UK biobank.

Appendix: First set of reviews: Point-by-point response to reviewer comments. Our response is indicated in blue and by indentation.

Graham et al.

Your manuscript entitled "Sex-specific and pleiotropic effects underlying kidney function identified from GWAS meta-analysis" has now been seen by 3 referees. You will see from their comments below that while they find your work of interest, some important points are raised. We are interested in the possibility of publishing your study in Nature Communications, but would like to consider your response to these concerns in the form of a revised manuscript before we make a final decision on publication.

Thank you for reviewing this manuscript and allowing us an opportunity to revise and resubmit a version improved by reviewers' and editors' expert opinions.

We therefore invite you to revise and resubmit your manuscript, taking into account the points raised. We would note that while we agree with Reviewer #2 that formal replication would be desirable, we acknowledge that replication cohort(s) of sufficient sample size and power might not be available and we would not insist on this particular point. We would, however, expect satisfactory responses to all other comments and to be more upfront about the current "replication" analysis being performed on related traits (and the associated limitations). Please highlight all changes in the manuscript text file.

Thank you. We will endeavor to address all reviewer points one-by-one below.

Please be aware that for certain types of new data, including most types of genetic data, journal policy is that deposition in a community-endorsed, public repository is generally mandatory prior to publication. Data submission can be a lengthy process, and we strongly suggest that you begin this well in advance of potential publication. Please include a statement about data availability in your point-by-point letter accompanying your revisions.

We will deposit summary statistics from the meta-analysis publicly. We will post results in this location (<http://csg.sph.umich.edu/willer/public/eGFR2018>) prior to publication and in the GWAS catalog.

Reviewer #1 (Remarks to the Author):

1. The sex differences in the incidence and progression of chronic kidney diseases have been examined in many experimental and clinical studies – the general consensus is that while sexual disparity exists, it is a male sex that shows greater susceptibility and faster progression of non-diabetic kidney disease than the opposite gender (please see for example J. Reckelhoff's and/or J. Neugarten's publications). The authors suggest the contrary in the introduction and their evidence is based on a reference to a book chapter. I think this requires a thorough re-examination.

We have updated the introduction to clarify this point and have added additional references (Carrero et al, Nature Rev Neph 2018; Reckelhoff and Sampson, Am J Physiol Regul Integr Comp Physiol 2015; Neugarten J Am Soc Nephrol. 2002)

2. The purpose and the details of prioritising genes within loci implicated in GWAS are not convincing. Initially, it appeared that this “functional prioritisation” was conducted to provide some input into DEPCIT. However, in Figure S7, the authors clearly demonstrate that DEPICT data was used actually as an input into the prioritisation analysis. It appears that the proximity of a gene to a sentinel variant was given the top priority in the “consensus” above the insights from DEPICT, eQTL studies and input from missense proxy SNP which is very surprising. There is a large body of evidence showing that the genes closest to the sentinel variant are frequently not the biological mediator of the identified associations. I think it would be sensible i) to report all the GWAS associations first using the gene closest to the sentinel SNP first (a common practice in GWAS), ii) conduct a separate “prioritisation” analysis to see how frequently the “prioritised” gene is different to the one implicated by proximity.

We have updated the prioritization analysis to reflect the most likely gene as those identified by all three methods (DEPICT, eQTL colocalization, and missense $r^2 > 0.8$); if none, then genes identified by two methods; if none, then genes identified by any method; otherwise we list the nearest gene only when the listed approaches do not identify any candidate genes.

3. The UK Biobank is a very sensibly selected resource for the replication of the GWAS signals but without some additional information and clarifications it is very difficult to assess the value of this analysis. Firstly, the description of the methods pertaining to this important analysis is inadequate. All the details of the selection, quality control filters for both variants and the phenotypes should be reported in detail. As is, it is not clear if the authors have derived their phenotypic information only from self-reported data or took advantage from linked Hospital statistics and mortality data. In the absence of blood biochemistry data, this could increase the number of “cases” and thus improve the power of the analysis. It also appears to me that certain phenotypes have been unnecessarily analysed separately – .i.e. “renal failure” and “renal dialysis”. I suggest that the authors take a full advantage of the data in UK Biobank but some thought should be given to the definition of “cases” and “controls”.

Phecode analyses are an automated process by necessity of the number of ICD codes available in a complete EHR. Although we have spent months assessing and deriving phecodes in collaboration with the leaders in this area (Denny etc. at Vanderbilt), some fine-tuning may still be beneficial. We carefully re-reviewed the 26 phenotypes we focused on and removed 3 that had broad definitions and higher numbers of cases than expected.

We have identified cases of CKD in the UK biobank using ICD-9 codes 585 and 586 and ICD-10 code N18. We have aimed to be thorough in our lookup of related traits, and so some phenotypes are a subset of others, for example individuals on renal dialysis are a subset of individuals having renal failure. The scheme of ICD-9 and ICD-10 phecodes is now in a preprint server and we now cite this reference for detailed descriptions of the phecodes (<https://doi.org/10.1101/462077>).

4. The details of eQTL analysis are shown in insufficient details both in the results and methods section. Firstly, it should be first appreciated that the kidney is very poorly represented in GTEx

– only about 30 samples exist with the RNA-seq-derived transcriptomic information and GWAS for the purpose of eQTL analysis. Were they included in this analysis? What were the other criteria of inclusion of a given tissue from GTEx in the analysis (normally only tissues with more than 100 samples are included). The kidney eQTL data come from two different sources – Ko’s paper used Tissue Cancer Genome Atlas resource (largely normal kidney tissue secured from cancer nephrectomies) while the recently published Gillies’ paper used kidney samples from patients with nephrotic syndrome. Did the authors examine each of these 2 kidney datasets separately and if yes, did the eQTL have to be demonstrated in both to satisfy the criteria of kidney-specific signal? The details of all eQTL analyses must be provided – what window was used, how the correction for multiple testing was calculated etc.

We are aware of the limitations of kidney eQTL data in GTEx and thus sought additional kidney eQTL datasets as noted by the reviewer. We have added additional information about the kidney eQTL analysis and criteria for significance in the methods. Also, we tested correlation between 111,739 SNP-eGene pairs common to our eQTLs and those that are publicly available from kidney cortex (unaffected parts of tumor nephrectomies). The Pearson correlation for GLOM-cortex and TI-cortex were 0.69 ($p < 2 \times 10^{-16}$) and 0.74 ($p < 2 \times 10^{-16}$), respectively. The vast majority showed consistent directional effect, albeit with effect-size heterogeneity.

5. The analysis of colocalisation between CKD and cardiovascular/metabolic traits in UK Biobank is interesting. There is clear redundancy among the selected phenotypes (i.e. essential hypertension and hypertension) and my comments above pertaining to UK Biobank-based analysis apply here as well.

Yes, however, the redundancy in phenotypes has a negligible impact on the multiple testing correction and, therefore, power. This is a standard PheWAS analysis.

6. The analysis of genetic risk scores clearly demonstrated that the variants associated with CKD in GWAS are effectively meaningless as predictive/diagnostic tools. It is only fair to fully acknowledge this message in the discussion in the context of other important insights that GWAS can provide.

It is unclear, at this time, if additional power from well-powered analyses of eGFR or CKD (using increased sample size and phenotype refinement as noted by the reviewer) may provide predictive utility of individuals at the tail of the risk distribution. Additional work is required in this area and we do not feel our study adequately provides evidence to claim there is no clinical utility of polygenic risk scores.

7. The last section of the results section – analysis of Hunt seems a little de-attached from the rest.

We have attempted to integrate these results better. Because HUNT is a new cohort where eGFR has not previously been analyzed (N=70k), we briefly describe the results in this cohort alone.

8. Page 9 – HUNT Study is a population-based resource. I am not clear why eGFR values were calculated in the individuals recruited into HUNT using MDRD equation and not CKD-EPI equation. Have the authors utilised the insights from biochemical analysis of blood to define the CKD status of the HUNT individuals (or was it based exclusively on ICD codes)?

CKD status was defined based on ICD codes.

9. Table 1 – it would be helpful to see beta and SE together with P-values for each of 4 cohorts that contributed to the meta-analysis.

We have added the study-specific results in supplementary table S1.

10. All tables require much more detailed legends.

These have been added.

Other comments:

11. That “eGFR levels below 60 mL/min/1.73m²...” (Page 4) characterise chronic kidney disease is not completely precise statement. Based on the currently used clinical definitions, CKD can be defined in patients with eGFR above this threshold in the presence of increased levels of ACR.

Thank you. We have corrected the text.

12. Diabetes and hypertension are not “chronic health conditions” (Page 4) but chronic diseases.

Thank you.

Reviewer #2 (Remarks to the Author):

The authors performed a meta-analysis on eGFR using two large published datasets of European ancestry individuals and of individuals from Japan, respectively, and included two additional smaller datasets of European ancestry.

Follow-up analyses included colocalization, lookup for different traits, and sex-stratified analyses. Although several new genetic associations with eGFR were revealed, a major drawback of this study is the lack of replication (except for the sex-stratified results). The alternatively conducted test for association of the index variants in kidney related traits in the UK Biobank addresses this problem only very limited because the tested endpoints differ from eGFR (including continuous traits vs. case/control status, renal failure vs. whole eGFR

spectrum).

We agree that replication would be important to the field going forward and we have entered discussions with Andrew Morris, Christian Pattaro and Christian Fuchsberger to enter into a joint meta-analysis with these groups in the future. However, the work we present also highlights new results from 96,329 newly analyzed individuals and a new focus on sex-specific effects. We have also attempted to replicate the sex-specific eGFR findings in sex-specific CKD estimates.

The authors state that genetic studies are needed to verify these associations in more diverse cohorts. Given that this study includes an almost equally sized sample set of EA individuals and individuals from Japan, it would be interesting to see the overlap and the differences in the eGFR associations between BBJ and the European ancestry studies which would be first step in this direction.

Thank you for this suggestion and we have added a short discussion of the differences between the Japanese and European ancestry results.

I have some additional comments and questions that should be addressed:

- Kidney-Specific eQTL Associations: As far as I understood, the significant eQTLs were constrained for not having eQTL in other tissue than kidney. The rationale for this approach is not quite clear. Variants that may have eQTLs across different tissues including kidney and would be excluded by this approach. Although this was a kidney specific eQTL analysis, it would be interesting to see also the results without removing loci that have eQTLs in non-kidney tissues.

Thank you. We have now identified eQTLs that are present in kidney tissue, and not necessarily specific to kidney tissue.

- for the DEPICT results a Bonferroni correction was applied for both tissue and geneset enrichment, which is very conservative given that in Tables S8 and S9 a FDR is provided. Thus, the authors report only the tip of the iceberg. A commonly used FDR cutoff of 5% would be more informative while controlling the type I error. Pattaro et al. (2016) reported already tissue enrichment beyond the urogenital tract that at least partially overlap with the results of this study using the proposed FDR cutoff. The authors should be consistent regarding the naming of the reported results: according to Supplementary Table 8 they report a mix of tissue names and MeSH terms of different levels for the DEPICT results, i.e. both significant tissue enrichments belong to the urogenital system.

We have now added more of the results by relaxing the threshold for inclusion to FDR 5%.

- prioritization in gene annotation: It seems not obvious that a missense variant in moderate LD ($r^2 > 0.3$) should be given a priority for gene selection. If the missense is really causal, I would expect a high LD with the lead variant, or the missense could be an independent variant which could be validated by a conditional analysis. I suggest the authors justify this criteria or increase the LD cut-off substantially. The prioritization by colocalization seems misleadingly named as eQTL in Suppl Figure 7. Finally, please provide the final selection criteria that was used to define a prioritized gene e.g. in Supplementary Table 1 and for the results of Table 2

Missense variants may not always be in high LD with the index variant if the frequencies are different (for example, R46L in PCSK9 has a frequency of 2.3% and is in r^2 of 0.03 with the initial lead variant at that locus:rs11206510 (Willer et al. 2008). However, we agree that most of the time, the 'causal' variant for an association signal will be either the lead SNP or in high LD with the lead SNP. We have removed Supplementary Figure 7, and have updated the Supplementary tables with the selection criteria for all prioritized genes. We now require the missense variant to be in $r^2 > 0.8$ with the lead variant to prioritize a gene.

- colocalization analyses of the lookup of traits in the UK Biobank: Which tissue was used for the colocalization analyses? Were the genome-wide or nominally significant associations in UK Biobank used as basis for the colocalization analysis? Did the colocalized genes per locus overlap with the eGFR colocalization results?

We only perform colocalization analysis for phenotype associations that reached our PheWAS significance threshold (requiring at least one variant in the ± 500 kb region around the eGFR index variant to have $p < 5 \times 10^{-8}$ for the tested UK Biobank trait). We are not sure how to interpret the reviewer's last sentence and wonder if the reviewer is asking whether the prioritized genes at each locus overlap with the gene identified by eQTL colocalization results? Since eQTL and DEPICT (which uses eQTL) was used to prioritize genes, the gene identification would not be independent.

- The GRS lookup with CKD was performed in the UK Biobank sample. How was CKD status defined in this UK Biobank sample? How many CKD cases and controls were available? Please provide the ROC curves of the reported AUCs as Supplemental Figures (at least for the best performing GRS).

CKD status was defined using PheCodes. We have added methods for this to the manuscript. ROC curves are now supplemental figures as well as % with CKD for each PRS bin.

- lookup of significant hits in the UK Biobank for kidney-related phenotypes (Supplementary Table 5): Please provide the effect estimates in addition to the p-values. How many cases and controls for each UK Biobank trait were included in the analyses? Were the effect directions for the CKD associations in line with the ones from eGFR? Were the effect directions for hypertension as expected with respect to eGFR? Scatterplots of the effect sizes of the eGFR

associations vs. selected UK Biobank traits would be informative.

We have added the number of cases and controls to Supplementary Table 10, and the direction for eGFR, CKD beta and se to Supplementary Table 5. We have added the scatter plots of eGFR effect size relative to hypertension effect and CKD effect as Supplementary Figure 3.

- MGI study methods: Please provide information about the time range of multiple creatinine measurements per individual that were used to average the analyzed eGFR. Furthermore, were unrelated EA individuals used (indicated in the reference publication)? Please add this information to the methods.

Added.

- sex-specific associations in the HUNT study: please specify which of the genes were prioritized as functional candidates based on colocalization and which ones on missense on high LD. Please provide also the effects of the lookup in UK Biobank in addition to the reported p-values.

We have clarified the source of evidence for biological candidate genes at each locus in new supplementary table 2.

Furthermore, the statement in the Methods "loci were identified separately in men and women and were filtered to those that were significant in one sex but near nominal significance in the other" is imprecise: which p-value exactly was defined as "near nominal"?

We have clarified this in the text ($P > 0.005$)

- Novel loci revealed in HUNT: The two reported novel loci for CKD are not plausible based on the very low MAF in combination with the number of cases, and without replication performed. A suitable MAF or minor allele count filter should be applied to the meta-analysis results, esp. for CKD. It should be mentioned, that the identified rare stop-gain variant in the known locus PKD2 still needs to be replicated in independent studies.

We have added minor allele counts and frequency within cases and controls to Supplementary Table 14 and have updated the text to point out the need for replication of these variants.

- Please provide gene names in Supplementary Tables 6, 11, 13. In addition, in Supplementary Table 13 it would be of interest to see whether prioritized genes of the conditionally independent signals differ from the prioritized genes of the main signals

We have added this information to the tables.

- Supplementary Table 1: please provide the in which order the datasets are listed in the "direction" column

Added.

- Discussion: The association of ABO with type 2 diabetes is rather weak compared to other phenotypes, therefore the postulated link of the ABO gene to CKD via type 2 diabetes is very speculative. Furthermore, given the design of this study (i.e. limiting the lookup in the UK Biobank to cardiovascular traits) and the small overlap of 6 out of 147 loci that colocalize with hypertension, I am not convinced that this study really "highlights" the pleiotropic associations with cardiovascular disease as stated. Therefore, this statement should be toned down.

Thank you, we have made this revision.

- Where any known hits not significant in the current meta-analysis analysis?

Two (of 127) previously reported variants were not significant in the current meta-analysis. These are given in supplementary table 6.

- limitations of this study should be stated, such as mix of population-based cohorts and disease cohorts (i.e. BBJ); no replication; heterogeneity in phenotype generation (different eGFR formulas and trait transformations)

Thank you for this suggestion.

- In addition to the lambdas provided in the QQ plots, LD score regression intercepts should be reported for the meta-analyses to disentangle possible inflation from polygenecity.

These are provided.

- Suppl Fig 6: There is a branch of p-values that deviates from the median line, i.e. where the p-value based meta-analysis p-value is higher than in the standard-error approach. Can the authors explain this deviation, i.e. why only a subset of associations is affected by this?

This is a set of SNPs from a locus in which the frequency differs between Japanese and European individuals, thus the square root of the sample size is not as effective at estimating the standard error when the allele frequency differs.

- Suppl Fig 6: Please rephrase "overwhelmingly" in the figure description, as a high correlation is rather expected for these type of analyses

Thank you for the careful review.

Reviewer #3 (Remarks to the Author):

The paper reports results from GWAS meta-analyses of eGFR that includes the HUNT study and MGI biobank, and two other GWAS/consortium data. They identified 147 loci including 53 novel loci. They also report sex-specific findings and associations of SNPs with multiple outcomes. There are several issues that need further clarifications. These include the strategy used for meta-analyses (minimum allele frequency, minimum number of samples contributing data), which can affect the number of identified associations, and the strategy for gene prioritization, which can affect downstream analyses. Sex-specific analyses should include interaction methods. Replication was performed using different phenotypes than those used for the discovery. Additional comments are shown below.

Introduction

2nd paragraph: CKD associations with lipids are inconsistent in studies and the statement needs to be supported by additional references.

We have added additional references in support of this.

3rd paragraph: the authors make an argument that HRC dense imputation enables discovery of new loci but a comparison of their results with imputed 1000 genomes reference panel is not included.

We have instead provided a reference that HRC imputation covers more of the genome than 1000 genomes. We feel that the effort involved in using and comparing to an older and less dense imputation reference panel is not the best science.

Results

Page 4, lines 97-98. The rationale for prioritization of genes for downstream analyses using the genes nearest to the index variant is not supported by functional evidence or the own authors' results shown in Table S4.

We only use the nearest gene if there is no other biological candidate available. Evidence suggests the nearest gene is the causal gene in ~50% of the loci.

Page 5, lines 105-108. These results are not replication as the phenotypes are disease-related and not eGFR.

We have modified the language to indicate we were examining whether variants involved in eGFR, a quantifiable measure of kidney function, are involved in chronic kidney disease. Although not a strict replication, this lends validity to the study and the eGFR association results reported.

Lines 111-114, this replication is not valid given the overlap in samples with published studies.

We have reported the association results for the current meta-analysis, which partially overlaps with previously reported studies, and for meta-analysis of HUNT and MGI alone which does not have overlap.

Lines 121-122. Specify if the eQTL is from normal kidney tissue or individuals with CKD, and if human or mouse tissue.

We have added some text to clarify these experimental details.

Page 6, lines 146-147, more details on the samples used for colocalization analysis including ancestry are needed.

We have added details to this section of the methods

Lines 155-172. How much of the eGFR heritability and variance are explained by the SNPs? The results suggest very little overall effects.

We have added an explanation of the heritability explained and how the loci can provide insight into biological underpinnings of disease, even if heritability or effect sizes are low. For example, variants in PCSK9 explain a small proportion of heritability of lipid phenotypes, but PCSK9-inhibitors have a dramatic effect on LDL cholesterol.

Line 168 statement related to differing lifestyle or hormonal factors needs to be supported by references.

We have added a reference for this.

Sex-specific analyses. Results may be driven by small samples within strata. A better strategy is using interaction methods.

We have also added this analysis. For the sex-specific loci, the variants were relatively common so the smallest stratum had between 6,895 and 7,911 individuals.

Page 7, lines 188-197, report sex-specific direction of effect for variants. Given different strategies for phenotype transformation across multiple studies, how this was compared to HUNT?.

P-value based meta-analysis was performed to look for consistently stronger associations in one sex than another. We have clarified this in the manuscript.

Lines 199-208. Should you used a more stringent p-value threshold given multiple traits tested?

We have used the standard genome-wide significance threshold of $p < 5 \times 10^{-8}$ for these analyses. We did not further adjust this threshold as each trait was analyzed separately.

Methods

Page 9, lines 263-264. HUNT study description. Is the study a longitudinal cohort or a multiple cross-sectional study?. How many individuals were recruited and how many were genotyped? Which criteria were used for selection of people for genotyping.

This is described in other publications and we have added the references. We have also added some details to the methods.

Line 280, adjustments for batch suggest that there were multiple genotyping and potential different studies combined within the HUNT study. Please clarify.

Because 70,000 samples from the HUNT study were genotyped, they were not genotyped in one enormous batch due to reagent batches etc.. Samples were randomized between batches and analyses were performed using batch as a covariate.

Page 10, lines 314-325. For the MGI, more details are needed on the visits used for serum creatinine, number of measures and timeframe selected for eGFR measures. In addition, more details are needed on relatedness and how this was accounted in the analyses.

We have added these details to the methods section.

Lines 327-330. There were also differences in eGFR equation estimation.

We have added a note of this difference to the discussion section.

Page 11, lines 334. The strategy for meta-analyses used results from at least 2 studies, so samples may be small for some of the findings, particularly for the variant with a MAF=0.01 (rs72683923). Please include the N for each variant in Table 1. Also highlight the prioritized strategy for genes for each SNP in this Table by including a column with the evidence. What was the filter for allele frequency in analyses?

We have edited supplementary table 1 to add this additional information and have clarified the methods section.

Line 343. The LD strategy for gene annotation and most significant variant within 1 Mb window are different, please clarify.

We used an LD r^2 cutoff of 0.2 for a pairwise comparison of the most significant index variants to ensure that regions with LD extending greater than 1 Mb were not mistakenly considered to be new loci. In annotating missense variants, we used an LD threshold of $r^2 > 0.8$ to identify all variants that may be in moderate LD with the index variant. We have included this in the methods section.

Line 355-258, include number of samples for each resource and if human or not. "Kidney association results" is a confusing term, please change to "eGFR associations".

Thank you, we have changed this term.

Page 12, lines 375-376, the studies chosen have differences in eGFR estimating equation that can affect the betas.

We have meta-analyzed only those studies which were inverse-normalized, so that betas are on a standardized scale.

Figure 2, the number of SNPs shown in the figure are less than 127 described in the legend, please clarify.

We have edited the figure legend to reflect that only variants with additional significant associations are plotted.

Table 1. Please include European and East Asian allele frequencies.

We have added these to supplementary table 1

Supplementary figure 3. QQ plots, the sex-specific lambda results are the same to the 6th decimal, please revise.

We have repeated this calculation and the values are correct as given. This is due to both analyses having the same median p-value.

Supplementary figures 4 and 5. Include lambdas.

We have added these

Results from supplementary figure 6 need to be included in the discussion and described in methods. Are the reported results based on p-values or standard deviations?

We have included a discussion of supplementary Figure 6 in the methods section, and have added a discussion of variants with differing p-values between the meta-analysis methods to the figure legend. The reported results are based on p-values.

Supplementary figure 7 is not needed.

We have removed this figure and have edited the supplementary tables to instead show the evidence for the gene prioritization.

Table S7. Include the source for eQTL. It looks like most are from GTEx tissues. Where are the results for the NephQTL and Ko et al.?

We have edited the supplementary tables to show the source of each tissue for the eQTL analysis.

Table S10. This information should be included in methods instead of a table.

As we have extended this table to include sample sizes and ICD-codes, we think it would be too lengthy to include in the methods section directly.

Reviewer #1 (Remarks to the Author):

There is an improvement in the quality or responses to the Reviewers' comments. However, a few points will require more attention.

1. My general comment is that each response should be followed by precise information on where exactly in the text and/or Supplementary materials the change was made (Page, lines etc).
2. It is still quite difficult to follow the data from cis-eQTL analysis. I suggest that the authors include a flowchart that illustrates all stages of this section of the paper including the proportion of their GWAS index variants in this analysis, the selection of appropriate datasets, the correction for multiple testing, the criteria for kidney-specificity, and the final outcomes.
3. Line 365: reads: "kidney specific eQTL results ...". This should read: "kidney eQTL results". Indeed, kidney specificity is established at a later stage of this analysis.
4. Lines 113-116: reads: "To identify variants that may be acting through regulation of gene expression within the kidneys, we selected eGFR index variants which were significant eQTLs ... "
I think the authors meant: "To identify variants that may be acting through regulation of gene expression within the kidneys, we examined which eGFR index variants were significant eQTLs..."
5. There is an additional published support for genes regulated by CKD eGWAS variants reported on Page 5 i.e. DPEP1, FGF5, MUC1. Please see Nat Commun. 2019;10:29 and Nat Commun. 2018;9:4800. This warrants an acknowledgment in the discussion.

Reviewer #2 (Remarks to the Author):

The authors now sufficiently addressed all my questions.

However, there is one additional issue that came up while reading answer 7 to reviewer #3: the variance of 25.3% explained by the GRS is likely substantially overestimated. The formula summing up $\beta^2(1-f)^2f$ for estimating the variance of the SNPs holds only for statistically independent SNPs, which is not the case for the applied $r^2 < 0.4$ threshold. I suggest removing the sentence stating the explained variance of the GRS from the results and methods as it provides minor information only, unless it is estimated more precisely e.g. using individual level data.

Reviewer #3 (Remarks to the Author):

The revised paper has greatly improved with authors addressing most of the concerns. Some modifications are still required:

1. Page 122-123, coloc methods of eQTL and GWAS summary statistics assumes the same ancestry for eQTL and GWAS, so results using European and East Asian should be revised. Add the number of samples that contributed eQTL from NephQTL and Ko et al to methods.
2. Question related to strategy used for meta-analyses. Table 1 shows that for variant rs35136921 only 177,995 of 350,504 participants contributed data and for variant rs4825261 only 96,329 individuals are included. Some of this information is currently in supplementary Table 1 and needs to be included in Table 1, column "direction".
3. Clarify if the allele frequencies reported in Table 1 are from European or East Asian or both
4. For Figure 3, include the number of women and men in the Hunt study. I could not find this information in the study description or Table S13.
5. For the new paragraph starting at page 212, the authors should comment on recent publications in Nature Medicine (Qiu et al) and Nature Communications (Morris et al) that use single-cell kidney gene expression.

Third set of reviews: Point-by-point response to reviewer comments. Our response is indicated in blue and by indentation.

Editor: We therefore invite you to revise and resubmit your manuscript, taking into account the points raised. We would also ask you to carefully go through the manuscript text again and avoid overly optimistic language around the value of the PRS and to moderate claims of "strong support" based on nominally significant associations. Please highlight all changes in the manuscript text file.

We have edited the paragraph with the PRS results, it now reads as:

"In summary, these results show that the variants identified from association studies of eGFR are correlated with the presence of CKD on a population level (**Supplementary Figure 4**). However, the results are not sufficient to identify individuals with CKD from those without. This is consistent with findings from prior studies examining GRS of eGFR²⁷."

We have edited the paragraph with the report of nominally significant findings in UK Biobank to add "initial", it now reads as:

"Instead, we tested for association of the index variants in kidney related traits in the UK Biobank (CKD, hypertensive CKD, renal failure, acute renal failure, renal failure NOS, renal dialysis, or other disorders of kidney and ureters). Seven of the 48 novel variants (5 were lost due to poor imputation) and 27 of the 85 lead variants in known loci that were available in the UK Biobank were at least nominally associated and had corresponding direction of effect with one or more UK Biobank kidney-related phenotypes, providing initial support for the biological validity of the eGFR results (**Supplementary Table 7, Supplementary Figure 2**)."

Reviewer #1 (Remarks to the Author):

There is an improvement in the quality of responses to the Reviewers' comments. However, a few points will require more attention.

Thank you. We are grateful for the detailed comments provided by the reviewers.

1. My general comment is that each response should be followed by precise information on where exactly in the text and/or Supplementary materials the change was made (Page, lines etc).

We apologize for not inserting line numbers. The editor said this was not necessary since we quote the revised text in the response to reviewers.

2. It is still quite difficult to follow the data from cis-eQTL analysis. I suggest that the authors include a flowchart that illustrates all stages of this section of the paper including the proportion of their GWAS index variants in this analysis, the selection of appropriate datasets, the correction for multiple testing, the criteria for kidney-specificity, and the final outcomes.

We use 3 different sources of eQTL results, and require significance for at least 1 tissue/gene to be reported. If eQTL association is observed in kidney-related tissues ($P < 6.7 \times 10^{-6}$) and not in other tissues ($P > 6.7 \times 10^{-6}$), then we deem it "kidney-specific". All GWAS

index variants (147) were used in analysis and significance was based on correction for testing of 147 index variants in 51 tissues types. We have added a flowchart as **Supplementary Figure 9** to help clarify this and point to this figure in the “eQTL Analysis” section of the methods (page 12)

3. Line 365: reads: “kidney specific eQTL results ...”. This should read: “kidney eQTL results”. Indeed, kidney specificity is established at a later stage of this analysis.

Thank you. We have made this correction.

4. Lines 113-116: reads: “To identify variants that may be acting through regulation of gene expression within the kidneys, we selected eGFR index variants which were significant eQTLs ... “

I think the authors meant: “To identify variants that may be acting through regulation of gene expression within the kidneys, we examined which eGFR index variants were significant eQTLs...”

Thank you for this edit.

5. There is an additional published support for genes regulated by CKD eGWAS variants reported on Page 5 i.e. DPEP1, FGF5, MUC1. Please see Nat Commun. 2019;10:29 and Nat Commun. 2018;9:4800. This warrants an acknowledgment in the discussion.

We have added references to the manuscripts which have been published since our initial submission. The text on page 8 now reads:

“Recent studies have further examined the role of gene expression in eGFR and CKD. Xu et al., performed Mendelian randomization analysis using gene expression data to identify a causal role for *MUC1* expression on eGFR⁴⁰. Single nucleus RNA-sequencing using cells from a human kidney donor identified expression of *DPEP1* that was specific to the proximal tubule⁴¹. Moreover, glomerular and tubular specific gene expression associations have been found to be significantly enriched for CKD and eGFR GWAS results⁴², emphasizing the need to consider eQTLs in kidney tissue when prioritizing genes from kidney-related GWAS.”

Reviewer #2 (Remarks to the Author):

The authors now sufficiently addressed all my questions.

However, there is one additional issue that came up while reading answer 7 to reviewer #3: the variance of 25.3% explained by the GRS is likely substantially overestimated. The formula summing up $\beta^2(1-f)2f$ for estimating the variance of the SNPs holds only for statistically independent SNPs, which is not the case for the applied $r^2 < 0.4$ threshold. I suggest removing the sentence stating the explained variance of the GRS from the results and methods as it

provides minor information only, unless it is estimated more precisely e.g. using individual level data.

We added the variance explained as specifically requested by reviewer #3 in review #1. We have repeated the analysis in a list of genome-wide significant, independent markers (defined by $r^2 < 0.2$ and $p\text{-value} < 5 \times 10^{-8}$) and added the result to the text (page 6): “The 1,189 variants included in this risk score explain an estimated 25.3% of the variance in eGFR levels, while the GRS constructed using only the significantly associated independent variants ($p\text{-value} < 5 \times 10^{-8}$, $r^2 < 0.2$) is estimated to explain 9.4% of the variance in eGFR.”

Reviewer #3 (Remarks to the Author):

The revised paper has greatly improved with authors addressing most of the concerns. Some modifications are still required:

1. Page 122-123, coloc methods of eQTL and GWAS summary statistics assumes the same ancestry for eQTL and GWAS, so results using European and East Asian should be revised. Add the number of samples that contributed eQTL from NephQTL and Ko et al to methods.

Nearly all eQTL results are in mixed ancestry samples, including GTEx (85% white) and NephQTL (52% European, 9% Asian). Ko et al. included only individuals of European ancestry. We felt it was more useful to perform colocalization methods, with the caveat that LD structures may not be exactly the same (but largely similar), which allows us to interpret evidence of colocalization as indicating the same signal, even if LD is slightly different. If there is not colocalization, reasons can be due to LD differences (unlikely between even European and East Asian as LD is largely similar) or different association signals. Since we do not highlight eQTL signals that do not colocalize with GWAS signals, we are already using this test in the most conservative manner. If we do not use the colocalization test at all (as hinted at by the reviewer) due to LD differences (since we do not have access to GTEx individual-level statistics to generate European eQTL results only), we will be including additional non-colocalized eQTL signals as candidate genes which may mislead future functional experiments.

We have added **Supplementary Figure 9**, which gives the sample sizes for the eQTL data from Ko et al and NephQTL. Sample sizes for all tissues (both kidney and those from GTEx) are given in **Supplementary Table 4**.

2. Question related to strategy used for meta-analyses. Table 1 shows that for variant rs35136921 only 177,995 of 350,504 participants contributed data and for variant rs4825261 only 96,329 individuals are included. Some of this information is currently in supplementary Table 1 and needs to be included in Table 1, column “direction”.

The N is already reported in Table 1. We believe that association in $> 90,000$ individuals and at least 2 studies is robust to avoid false positives. Since all information requested by the reviewer (including direction for each individual study) is available in Supplementary Table 1, we are not sure what the reviewer is asking for here.

3. Clarify if the allele frequencies reported in Table 1 are from European or East Asian or both

We have edited the Table legend to read “^aReported frequency and direction of effect is with respect to the alternate allele for the combined meta-analysis. Study specific associations and allele frequencies are given in **Supplementary Table 1.**”

4. For Figure 3, include the number of women and men in the Hunt study. I could not find this information in the study description or Table S13.

We have added this to Figure 3 and Supplementary Table 13.

5. For the new paragraph starting at page 212, the authors should comment on recent publications in Nature Medicine (Qiu et al) and Nature Communications (Morris et al) that use single-cell kidney gene expression.

We have added references to these manuscripts that were published after initial submission. This section of the text now reads (page 8):

“Recent studies have further examined the role of gene expression in eGFR and CKD. Xu et al., performed Mendelian randomization analysis using gene expression data to identify a causal role for *MUC1* expression on eGFR⁴⁰. Single nucleus RNA-sequencing using cells from a human kidney donor identified expression of *DPEP1* that was specific to the proximal tubule⁴¹. Moreover, glomerular and tubular specific gene expression associations have been found to be significantly enriched for CKD and eGFR GWAS results⁴², emphasizing the need to consider eQTLs in kidney tissue when prioritizing genes from kidney-related GWAS.”

Appendix 1: Second set of reviews: Point-by-point response to reviewer comments. Our response is indicated in blue and by indentation.

Reviewer #1 (Remarks to the Author):

1) The importance of gender as a risk factor for chronic kidney disease requires a more balanced interpretation. Not all studies reported so far showed the higher prevalence of chronic kidney disease in women than men. For example, data from Japan show clear male disadvantage in both age-related decline in kidney function and incidence of end-stage renal disease (Kidney International 2008;74:505-512 and Kidney International 1996;49:800-805).

We have removed this portion of the introduction to instead focus on CKD overall.

2) Neither Table S2 nor the text on Page 5 specifies the tissue(s) in which the eQTL analyses were conducted. I suspect these are GTEX non-kidney tissue? If the kidney resources were used in this analysis how the discrepancy between the kidney and non-kidney tissue was interpreted?

As previously requested by reviewer 2, we have included eQTL results that are significant in non-kidney tissues as well as kidney tissues. The tissue and gene for the eQTL analysis are given in Table S5, as requested by reviewer 3. We have now also updated Table S2 to specify which eQTL results are from kidney tissue and the text on Page 5 to specify that both kidney and non-kidney tissue eQTL results were included.

3) UK Biobank analyses are in essence based on data generated by previously conducted genome-wide association studies conducted in UK Biobank. Did the authors apply any quality control filters at i) genotype, ii) participant, iii) phenotype-level prior to their analysis of association between eGFR variants and kidney/cardiovascular phenotypes?

Yes, we did. We have added more detail about the genotype, individual and phenotype-level exclusions in the supplemental methods. The new text reads:

“We performed analysis using individuals in the white British subset of UK Biobank that were included in the kinship calculation, excluding those identified as outliers based on the missingness rate and heterozygosity, and those missing from the UK Biobank phasing calculations. At the genotype level, we included all variants used for calculation of the kinship matrix and excluded variants after imputation with INFO score < 0.3 and variants not in the HRC imputation panel.”

4) How many variants of 53 novel CKD variants were available for look-up in UK Biobank? (On page 5 it reads “Thirteen of the 48 novel variants...”)

We apologize for the lack of clarity. We have now revised the sentence to be more clear. It now reads as “Seven of the 48 novel variants (5 were lost due to poor imputation)...”.

The number of variants has changed as we have now considered direction of effect in addition to p-value.

5) There is lack of consistency in reporting the results of this analysis in Table S7. For “CKD chronic renal failure” beta, SE and P-values are reported, for other traits, just P-values. Why?

We apologize for this omission. For all traits, p-values, beta, and se are now reported in Supplementary Table 7.

6) I do not think that the nominal significance level is right for the analyses in UK Biobank given the number of tests conducted. There must be some penalty for multiple testing. Had the authors triangulated individual-level data from multiple sources of information (self-reported, hospital statistics, mortality etc.) available in UK Biobank into one over-arching phenotype i.e. “history of chronic kidney disease”, they would have had higher number of disease cases for “replication” purposes (and thus better power), “purer” controls, fewer phenotypes and less significant requirement of penalty for multiple testing.

We clearly state in the manuscript which additional phenotypes reach the more stringent genome-wide significance threshold, and which reach only nominal significance. We feel important to provide this information in a supplemental table to delve deeper into patterns of phenotypic heterogeneity across all loci. Text from manuscript: “A PheWAS analysis of the eGFR-index variants across 23 cardiovascular and diabetes-related phenotypes in UK Biobank, excluding individuals with CKD, identified 7 phenotypes for which a subset of the index variants was also significant (p-value < 1.48×10^{-5} , Bonferroni correction for 23 phenotypes and 147 index variants): diabetes, coronary atherosclerosis, hypertension, essential hypertension, pulmonary heart disease, phlebitis and thrombophlebitis, and ischemic heart disease (**Supplementary Tables 7,12, Figure 2**). Colocalization analysis with these phenotypes identified 7 loci (prioritized genes: *FGF5*, *PRKAG2*, *TRIB1*, *DCDC5/MPPED2*, *L2HGDH/SOS2*, *UMOD*, *SALL1*) having significantly colocalized association signals with hypertension, essential hypertension, and/or coronary atherosclerosis and 1 locus (prioritized gene: *GCKR*) that colocalized with association of type 2 diabetes (**Supplementary Table 7**). Six of the seven index variants within loci that showed significant colocalization with the cardiovascular traits were associated with essential hypertension and/or hypertension, underscoring the connection between high blood pressure and CKD. In addition, the index variants were examined for association with 1,400 traits phenome-wide (without exclusion of CKD cases). As shown in **Figure 2**, the index variants are significantly associated (p-value < 5×10^{-8}) with additional traits including hypothyroidism and lipid metabolism disorders.”

All individuals with mortality data attributed to either CKD or renal failure were included in the original phenotypes. We examined the impact of inclusion of self-reported kidney disease (questions were history of: 1) renal failure, 2) renal failure requiring dialysis, 3) renal failure not requiring dialysis) and found that the overarching phenotype of “history of renal failure” resulted in 237 additional cases, a 3.4% increase relative to the previous definition of renal failure based on ICD codes (N=6,985). We felt that the nominal increase, as well as uncertainty of a self-reported phenotype in the absence of an ICD-code, did not warrant an entire new GWAS.

7) Responses to my questions about the eQTL analysis are not sufficient (comment 4). The authors should answer every question asked in this section and the answers should also be incorporated in the manuscript.

R1 original comment #4. The details of eQTL analysis are shown in insufficient details both in the results and methods section. Firstly, it should be first appreciated that the kidney is very poorly represented in GTEx – only about 30 samples exist with the RNA-seq-derived transcriptomic information and GWAS for the purpose of eQTL analysis. Were they included in this analysis? What were the other criteria of inclusion of a given tissue from GTEx in the analysis (normally only tissues with more than 100 samples are included). The kidney eQTL data come from two different sources – Ko’s paper used Tissue Cancer Genome Atlas resource (largely normal kidney tissue secured from cancer nephrectomies) while the recently published Gillies’ paper used kidney samples from patients with nephrotic syndrome. **Did the authors examine each of these 2 kidney datasets separately and if yes, did the eQTL have to be demonstrated in both to satisfy the criteria of kidney-specific signal? The details of all eQTL analyses must be provided – what window was used, how the correction for multiple testing was calculated etc.**

Original response: We are aware of the limitations of kidney eQTL data in GTEx and thus sought additional kidney eQTL datasets as noted by the reviewer. We have added additional information about the kidney eQTL analysis and criteria for significance in the methods. New text:

New response: We apologize for the lack of detail previously added. We considered kidney eQTL results to be those found in either the NephQTL and Ko et al. datasets and did not require significance in both. We have added additional text to state that these were used separately in the methods section. As also described in the methods section, we used a Bonferroni corrected significance threshold (6.7×10^{-6}) for testing of individual variants and a stricter threshold of 5×10^{-8} within a ± 500 kb window around the eGFR index variant when performing colocalization analysis. This paragraph now reads as:

“Publicly available eQTL association datasets from GTEx V7²⁴, NephQTL²³, and Ko et al.²² were each used separately to identify overlap between gene expression and identified eGFR association results. Specific tissue types, sample sizes, and links to public datasets included in the analysis are given in **Supplementary Table 4**. Kidney specific eQTL results were taken from only NephQTL and the Ko et al. datasets due to the small sample size of the GTEx kidney dataset. NephQTL includes kidney samples from individuals with nephrotic syndrome and the Ko et al. dataset includes normal kidney samples from the Cancer Genome Atlas (TCGA). Lookup of individual variants for association with gene expression was performed using a Bonferroni-corrected p-value threshold of 6.7×10^{-6} (correction for 51 tissue types and 147 index variants). The resulting associations were considered to be kidney-specific if an index variant was significantly-associated with expression of a given gene in any of the kidney-specific datasets but not in other tissues available (from GTEx). Colocalization analysis was performed using the R package coloc⁶⁴. Priors for p1, p2, and p12 within the coloc analysis were set to 1×10^{-4} , 1×10^{-4} , and 1×10^{-6} , respectively. Variants in the ± 500 kb region surrounding each eGFR

index variant were used for input into coloc. Within this region, we required at least one genome-wide significant (p -value $< 5 \times 10^{-8}$) eQTL variant prior to testing for colocalization. Following the criteria published by Giambartolomei et al.⁶⁴, eQTLs were considered to colocalize with the eGFR association results if the posterior probability (PP) for a shared variant was $> 80\%$. “

8) The authors appear to have ignored my request to contemplate the meaning of their analysis on genetic risk scores in the discussion.

The original request was to “acknowledge .. in the discussion” that “the variants associated with CKD in GWAS are effectively meaningless as predictive/diagnostic tools”. We respectfully disagree and explained why in our initial response and here again. AUC does not completely capture the predictive value of the PRS, particularly at the tails of the distribution. We have included a figure (Supplementary Figure 4) demonstrating the prevalence of CKD based on the PRS centile to provide evidence of our claim.

Supplementary Figure 4 legend excerpt: “CKD prevalence in the highest GRS percentile (1.05%) was 2.5 times higher than that of the lowest GRS percentile (0.43%) and 1.6 times higher than CKD prevalence among all other percentiles (0.65%).”

We believe that the results section sufficiently explains this, and the section relevant to this point currently reads: “In summary, these results show that the variants identified from association studies of eGFR are correlated with the presence of CKD on a population level and may be used to identify the individuals at highest risk (**Supplementary Figure 4**). However, the results are not sufficient to identify individuals with CKD from those without. This is consistent with findings from prior studies examining GRS of eGFR²⁷”

9) The authors have not responded explicitly to my question 8. MDRD equation is not the best choice to calculate eGFR in the population-based resource.

We apologize, this response appears to have been lost during edits of our response to reviewers last round. Because we performed inverse-normal transformation of residuals adjusted for covariates (age, sex, batch), we expected that using MDRD or CKD-EPI equation would have little impact on the phenotype used for association analysis. Indeed, we repeated the eGFR using CKD-EPI and found the correlation in the inverse-normal transformed residuals adjusted for covariates to be highly correlated ($r^2 = 0.995$). We have added the Figure below as Supplemental Figure 6 and added the following text to the methods “We also calculated eGFR using the CKD-EPI equation and after adjustment for covariates including age, sex, and batch followed by inverse normal transformation, the resultant eGFR phenotype values were highly correlated with those derived in the same manner after eGFR was calculated using MDRD (Supplementary Figure 6, $r^2 = 0.995$).”

10) I am not clear why information derived from serum creatinine (that is clearly available) was not used in combination with ICD codes to define better the CKD status of HUNT individuals. In any case, I do not see much value in the last section of the results devoted exclusively to the HUNT study given that these subjects were already included in the meta-analysis of eGFR (first section of the manuscript), my reservations pertaining to CKD definition in this study and the level of novel discovery in particular in the absence of replication.

We have removed this paragraph. Since we are authors from the HUNT study and this is our first publication on the HUNT study for kidney-related phenotypes, we had hoped to describe this large resource (N~70,000) but we defer to the reviewer and we now only describe results from the meta-analysis.

Reviewer #2 (Remarks to the Author):

In the revised version, most of my concerns were addressed or clarified, but some issues remain.

According to the conditional analyses methods and the Manhattan and QQ plots (Supplementary Figure 1), a MAF>0.5% filter was applied for the eGFR GWAS in HUNT. Why was no filter applied for the CKD analyses in HUNT? The new CKD associations are not convincing given the very low MAF and MAC. Besides the required replication (which was also stated by the authors), it is likely that these results are rather statistical artefacts of the association model which would also explain the implausible large effect sizes.

We suspect that the reason we identify lower frequency results in HUNT is because previous GWAS studies have only examined variants above a certain MAF threshold. Since the reviewer is skeptical of the findings with lower MAF and we cannot replicate these results in the short time frame, we have removed discussion of these results from the manuscript and include them only in the supplementary information.

The effect estimates for the association with CKD were added to Supplementary Table 5, but the effect estimates for the other traits are still missing. For hypertension, this information would be particularly helpful for interpreting the Supplementary Figure 2, i.e. which SNPs have a significantly positive effect on both increased eGFR and increased risk of hypertension. For all other traits listed in Supplementary Table 7, at least the effect direction should be provided (if the table becomes too complex by adding beta/SE) for better interpretation of both the pleiotropic effects and the colocalization results.

In addition, the description of Supplementary Figure 2 should include what the error bars represent (I assume the SE of the beta instead of e.g. a 95% CI).

We have added these effect estimates (beta, se, direction) for other phenotypes to Supplementary Table 5. We apologize for the oversight. We have added to Supplementary Figure 2 a description of the error bars: *"Error bars represent ± 1 SE."*

The time range (in years) spanning the multiple creatinine measurements in the MGI study is still not provided, or were the measurements taken within a few days only? This information would be helpful to get an impression of possible declining effects of eGFR given that the mean value was used for the analysis.

We have edited the text to report that the median time range between the first and last creatinine measurement was 2.4 years.

Thanks for clarifying the trait colocalization with the UK Biobank traits and solving my misunderstanding of the analyses performed. I suggest to state in the paragraph "Overlap with Related Traits" more clearly that trait-based colocalization between eGFR and UK Biobank traits were performed, to avoid confusion with eQTL-based colocalization described in the preceding paragraph.

Thank you for this suggestion. We have revised the heading.

As reported in the revised version, the $\lambda=1.20$ of the HUNT eGFR GWAS is high (Suppl Figure 6). An LD score regression for the HUNT GWAS results should be provided, too.

This is now provided. The LDSC intercept was 1.098.

Supplementary Tables 8 and 14: what does "snpeff ensemble summary" (column header) mean? What do the numbers in brackets in this column stand for?

Annotations were performed using WGS, and we report the SnpEff variant annotations. The numbers in brackets are the number of transcripts and number of affected transcripts for each gene and variant annotation, respectively. We have added descriptions of this to the Supplementary Tables. E.g. Supplementary Table 2 footnote: *"Annotation of variants was done using the SnpEff ensemble summary from WGS. The numbers in brackets denote the number of transcripts and number of affected transcripts for each gene and annotation type, respectively."*

Although effect alleles are provided in Supplementary Table 9 (eQTL), the corresponding effect directions are missing.

We have now provided the direction. We apologize for this oversight.

As far as I see, the question of Reviewer #1 regarding the calculation of eGFR in HUNT by the MDRD instead of the CKD-EPI formula was missed to answer.

Thank you. We have responded to reviewer 1 above.

Reviewer #3 (Remarks to the Author):

The authors have answered some but not all questions of this reviewer and additional clarifications are needed. The authors should restrain from making strong assertions that are beyond the scope of the findings, given their analyses are based on associations and predictive models.

Issues related to new text/data included in the revision:

1. A justification for a new co-author added to the paper is needed.

Because our first author is on maternity leave, an additional author was recruited to help address revisions.

2. References added to justify sex-specific effects are 3 review articles, one of them related to cardiovascular disease. Original research related to the prevalence of CKD in women is

preferable.

We have removed the discussion of differences in CKD prevalence between sexes from the introduction and instead focus on the prevalence of CKD overall.

3. I am not sure we know if CKD progression is faster in men versus women. This is a strong statement and it should be supported by references from original articles in humans.

We have removed this statement from the manuscript

4. The sentence "Several other health conditions interact with kidney function." does not make sense.

Thank you, we have revised this sentence. It now reads: "*Several other health conditions affect kidney function. Chronic diseases such as diabetes and hypertension directly influence the development of CKD, with environmental factors such as smoking accelerating disease progression⁵.*"

5. Results: LD score regression estimation of heritability, which population was used for this estimation, European or Japanese?

We have clarified this in the text to state that the European 1000 Genomes reference panel was used. We repeated the analysis using LD estimated from either 1) East Asian 1000G samples or 2) European 1000G and found similar results; 1) East Asian: 7.1% and 2) European: 7.6%.

6. Results: 13 novel variants had nominal replication in UK Biobank is NOT a strong supportive evidence for the biological validity of the eGFR findings. The results are based on p-values. At least a comparison on consistency in the direction of the association is needed.

We have edited the text to include only those results which were directionally consistent with kidney outcome (ie. Decreasing eGFR association and increasing kidney disease association) and at least nominally significant, and have removed the term "strong". The text now reads "*Seven of the 48 novel variants (5 were lost due to poor imputation) and 27 of the 85 lead variants in known loci that were available in the UK Biobank were at least nominally associated and had corresponding direction of effect with one or more UK Biobank kidney-related phenotypes, providing support for the biological validity of the eGFR results (Supplementary Table 7, Supplementary Figure 2).*"

7. Genetic risk scores explain 25% of eGFR variance in which dataset?

We have included in the methods section a statement that the variance explained by the risk scores was calculated using the effect sizes and frequencies from meta-analysis of BioBank Japan, MGI, and HUNT. The variance explained was estimated using $2pq \cdot \beta^2$ where p is the minor allele frequency, q is the major allele frequency and β is the per-allele effect estimate. The text reads: *“The proportion of variance explained by the GRS was calculated as the sum of $\beta^2(1-f)2f$ across all variants included in the risk score, where β and f are the effect size and frequency from meta-analysis of BioBank Japan, MGI, and HUNT. GRS were then calculated within UK Biobank as the sum of risk alleles carried by each individual weighted by the effect size of each variant.”*

8. Statement related to results from ROC that have a slightly better prediction of CKD in women DO NOT support use of genetic risk to identify individuals at risk.

We did not intend to say that the differences in prediction by sex support the use of GRS, but rather that GRS in general may be used to stratify individuals by risk level (Supplementary Figure 4). We have edited this sentence to clarify this point. The text reads:

“We also tested prediction of CKD using the best-performing GRS from the overall meta-analysis (without birth year or additional risk factors) separately in men and women. The GRS was slightly more predictive in women (AUC: 0.552) than in men (AUC: 0.538), possibly due to differing lifestyle or hormonal factors³⁴ between the sexes influencing the development of CKD. In summary, these results show that the variants identified from association studies of eGFR are correlated with the presence of CKD on a population level and may be used to identify the most at risk individuals (Supplementary Figure 4).”

Prior questions:

9. Question related to strategy used for meta-analyses (minimum allele frequency, minimum number of samples contributing data) was not answered.

We have included in the methods section a statement that we did not apply a pre- meta-analysis allele frequency or sample size filter. The relevant sentence is: *“Summary statistics from contributing studies were GC corrected prior to meta-analysis and were not filtered by minor allele frequency or sample size.”*

10. The literature on lipids and CKD is so far not supportive of association in humans. The citations are two review articles and an original study relating lipids to cardiovascular disease. Not sure why the authors want to stress lipids since these are not related to main findings.

We agree that the association of kidney disease with lipids is outside of the scope of this manuscript and have removed this sentence.

11. Clearly state in the main text the % of genes prioritize using the different strategies and % that was nearby gene.

We have added this to the results section: *“We were able to prioritize genes using these annotations for 126 of the 147 loci (86%). Loci that were not able to be prioritized through these methods were annotated as the nearest gene (21/147 loci, 14%).”*

12. The sentence “Of the 127 previously reported index variants for eGFR, 125 were at least nominally significant in the present meta-analysis”. As mentioned before, your meta-analyses include prior studies used for discovery so these results should not be reported.

We now only report results excluding previously published datasets: *“Excluding the previously published datasets, 56 of the 118 available variants were at least nominally significant in meta-analysis of HUNT and MGI alone (Supplementary Table 8).”*

13. Question related to kidney eQTLs from normal versus patients with nephrotic syndrome was not addressed. In the paragraph reporting eQTLs (Supplementary Table 9), it is still not clear which genes were eQTL for kidney cortex, tubulointerstitium or other. The source of the kidney eQTL is also not listed in the main text or Supplementary Table 9. Given nephrotic patients have CKD, the evidence for eQTL based on that data is less supportive for eGFR findings not related to CKD. Include the source of the kidney tissue by the genes in the main text and in table S9 (new column – normal versus nephrotic tissue). Include the paragraph where the new changes were done in the main document in your answer.

We had previously added Supplementary Table 4 which provides the samples sizes and data sources for all eQTL datasets (both kidney and non-kidney). This is noted at the top of Supplementary Table 9. We have now additionally added a column to Supplementary Table 9 noting the source of the eQTL data and for kidney tissues whether the tissue is from normal or nephrotic kidney samples. This paragraph in the main text has been edited to read: *“This identified 16 genes whose expression was associated with the eGFR index variants in kidney tissues, including 7 genes identified from normal kidney cortex tissue samples²² and 10 genes identified from kidney glomerulus or tubulointerstitium samples²³ from individuals with nephrotic syndrome (1 gene overlapped both datasets, Supplementary Table 9).”*

14. Phewas analyses need to account for the multiple test performed to avoid spurious associations.

We had previously used a genome-wide significance threshold of 5×10^{-8} when identifying pleiotropic associations, but have now edited the analysis to use a Bonferroni-corrected threshold of $p\text{-value} < 1.48 \times 10^{-5}$ (correction for 127 variants and 1400 traits).

15. Sex-specific analysis strategy is unconventional. The authors first tested genome-wide differences between men and women using a $p < 0.005$ for significance, then tested gene-sex interaction just for the three variants identified in the sex-stratified analyses. Follow-up analyses in the UK Biobank reporting associations with chest pain: this is a soft outcome, and the authors should examine the association with cardiovascular disease events instead.

We have changed the sex-specific analysis strategy to the reviewer’s preferred approach and have now analyzed all eGFR meta-analysis index variants using interaction tests. Due to the larger number of loci included and the change to eGFR meta-analysis index variants rather than those from HUNT, only one locus remains significant after Bonferroni correction for the 147 index variants. The association with chest pain is therefore removed from the text.

16. Question related to the ancestry used for colocalization analyses not answered.

Colocalization analysis was performed using the entire eGFR meta-analysis results (ie. 59% European and 41% East Asian individuals) with the white British subset of UK Biobank used for GWAS analysis of related traits. The PheWAS analysis was pre-calculated with white British individuals from UK biobank.

Appendix 2: First set of reviews: Point-by-point response to reviewer comments. Our response is indicated in blue and by indentation.

Graham et al.

Your manuscript entitled "Sex-specific and pleiotropic effects underlying kidney function identified from GWAS meta-analysis" has now been seen by 3 referees. You will see from their comments below that while they find your work of interest, some important points are raised. We are interested in the possibility of publishing your study in Nature Communications, but would like to consider your response to these concerns in the form of a revised manuscript before we make a final decision on publication.

Thank you for reviewing this manuscript and allowing us an opportunity to revise and resubmit a version improved by reviewers' and editors' expert opinions.

We therefore invite you to revise and resubmit your manuscript, taking into account the points raised. We would note that while we agree with Reviewer #2 that formal replication would be desirable, we acknowledge that replication cohort(s) of sufficient sample size and power might not be available and we would not insist on this particular point. We would, however, expect satisfactory responses to all other comments and to be more upfront about the current "replication" analysis being performed on related traits (and the associated limitations). Please highlight all changes in the manuscript text file.

Thank you. We will endeavor to address all reviewer points one-by-one below.

Please be aware that for certain types of new data, including most types of genetic data, journal policy is that deposition in a community-endorsed, public repository is generally mandatory prior to publication. Data submission can be a lengthy process, and we strongly suggest that you begin this well in advance of potential publication. Please include a statement about data availability in your point-by-point letter accompanying your revisions.

We will deposit summary statistics from the meta-analysis publicly. We will post results in this location (<http://csg.sph.umich.edu/willer/public/eGFR2018>) prior to publication and in the GWAS catalog.

Reviewer #1 (Remarks to the Author):

1. The sex differences in the incidence and progression of chronic kidney diseases have been examined in many experimental and clinical studies – the general consensus is that while sexual disparity exists, it is a male sex that shows greater susceptibility and faster progression of non-diabetic kidney disease than the opposite gender (please see for example J. Reckelhoff's and/or J. Neugarten's publications). The authors suggest the contrary in the introduction and their evidence is based on a reference to a book chapter. I think this requires a thorough re-examination.

We have updated the introduction to clarify this point and have added additional references (Carrero et al, Nature Rev Neph 2018; Reckelhoff and Sampson, Am J Physiol Regul Integr Comp Physiol 2015; Neugarten J Am Soc Nephrol. 2002)

2. The purpose and the details of prioritising genes within loci implicated in GWAS are not convincing. Initially, it appeared that this “functional prioritisation” was conducted to provide some input into DEPCIT. However, in Figure S7, the authors clearly demonstrate that DEPICT data was used actually as an input into the prioritisation analysis. It appears that the proximity of a gene to a sentinel variant was given the top priority in the “consensus” above the insights from DEPICT, eQTL studies and input from missense proxy SNP which is very surprising. There is a large body of evidence showing that the genes closest to the sentinel variant are frequently not the biological mediator of the identified associations. I think it would be sensible i) to report all the GWAS associations first using the gene closest to the sentinel SNP first (a common practice in GWAS), ii) conduct a separate “prioritisation” analysis to see how frequently the “prioritised” gene is different to the one implicated by proximity.

We have updated the prioritization analysis to reflect the most likely gene as those identified by all three methods (DEPICT, eQTL colocalization, and missense $r^2 > 0.8$); if none, then genes identified by two methods; if none, then genes identified by any method; otherwise we list the nearest gene only when the listed approaches do not identify any candidate genes.

3. The UK Biobank is a very sensibly selected resource for the replication of the GWAS signals but without some additional information and clarifications it is very difficult to assess the value of this analysis. Firstly, the description of the methods pertaining to this important analysis is inadequate. All the details of the selection, quality control filters for both variants and the phenotypes should be reported in detail. As is, it is not clear if the authors have derived their phenotypic information only from self-reported data or took advantage from linked Hospital statistics and mortality data. In the absence of blood biochemistry data, this could increase the number of “cases” and thus improve the power of the analysis. It also appears to me that certain phenotypes have been unnecessarily analysed separately – .i.e. “renal failure” and “renal dialysis”. I suggest that the authors take a full advantage of the data in UK Biobank but some thought should be given to the definition of “cases” and “controls”.

Phecode analyses are an automated process by necessity of the number of ICD codes available in a complete EHR. Although we have spent months assessing and deriving phecodes in collaboration with the leaders in this area (Denny etc. at Vanderbilt), some fine-tuning may still be beneficial. We carefully re-reviewed the 26 phenotypes we focused on and removed 3 that had broad definitions and higher numbers of cases than expected.

We have identified cases of CKD in the UK biobank using ICD-9 codes 585 and 586 and ICD-10 code N18. We have aimed to be thorough in our lookup of related traits, and so some phenotypes are a subset of others, for example individuals on renal dialysis are a subset of individuals having renal failure. The scheme of ICD-9 and ICD-10 phecodes is now in a preprint server and we now cite this reference for detailed descriptions of the phecodes (<https://doi.org/10.1101/462077>).

4. The details of eQTL analysis are shown in insufficient details both in the results and methods section. Firstly, it should be first appreciated that the kidney is very poorly represented in GTEx

– only about 30 samples exist with the RNA-seq-derived transcriptomic information and GWAS for the purpose of eQTL analysis. Were they included in this analysis? What were the other criteria of inclusion of a given tissue from GTEx in the analysis (normally only tissues with more than 100 samples are included). The kidney eQTL data come from two different sources – Ko's paper used Tissue Cancer Genome Atlas resource (largely normal kidney tissue secured from cancer nephrectomies) while the recently published Gillies' paper used kidney samples from patients with nephrotic syndrome. Did the authors examine each of these 2 kidney datasets separately and if yes, did the eQTL have to be demonstrated in both to satisfy the criteria of kidney-specific signal? The details of all eQTL analyses must be provided – what window was used, how the correction for multiple testing was calculated etc.

We are aware of the limitations of kidney eQTL data in GTEx and thus sought additional kidney eQTL datasets as noted by the reviewer. We have added additional information about the kidney eQTL analysis and criteria for significance in the methods. Also, we tested correlation between 111,739 SNP-eGene pairs common to our eQTLs and those that are publicly available from kidney cortex (unaffected parts of tumor nephrectomies). The Pearson correlation for GLOM-cortex and TI-cortex were 0.69 ($p < 2 \times 10^{-16}$) and 0.74 ($p < 2 \times 10^{-16}$), respectively. The vast majority showed consistent directional effect, albeit with effect-size heterogeneity.

5. The analysis of colocalisation between CKD and cardiovascular/metabolic traits in UK Biobank is interesting. There is clear redundancy among the selected phenotypes (i.e. essential hypertension and hypertension) and my comments above pertaining to UK Biobank-based analysis apply here as well.

Yes, however, the redundancy in phenotypes has a negligible impact on the multiple testing correction and, therefore, power. This is a standard PheWAS analysis.

6. The analysis of genetic risk scores clearly demonstrated that the variants associated with CKD in GWAS are effectively meaningless as predictive/diagnostic tools. It is only fair to fully acknowledge this message in the discussion in the context of other important insights that GWAS can provide.

It is unclear, at this time, if additional power from well-powered analyses of eGFR or CKD (using increased sample size and phenotype refinement as noted by the reviewer) may provide predictive utility of individuals at the tail of the risk distribution. Additional work is required in this area and we do not feel our study adequately provides evidence to claim there is no clinical utility of polygenic risk scores.

7. The last section of the results section – analysis of Hunt seems a little de-attached from the rest.

We have attempted to integrate these results better. Because HUNT is a new cohort where eGFR has not previously been analyzed (N=70k), we briefly describe the results in this cohort alone.

8. Page 9 – HUNT Study is a population-based resource. I am not clear why eGFR values were calculated in the individuals recruited into HUNT using MDRD equation and not CKD-EPI equation. Have the authors utilised the insights from biochemical analysis of blood to define the CKD status of the HUNT individuals (or was it based exclusively on ICD codes)?

CKD status was defined based on ICD codes.

9. Table 1 – it would be helpful to see beta and SE together with P-values for each of 4 cohorts that contributed to the meta-analysis.

We have added the study-specific results in supplementary table S1.

10. All tables require much more detailed legends.

These have been added.

Other comments:

11. That “eGFR levels below 60 mL/min/1.73m²...” (Page 4) characterise chronic kidney disease is not completely precise statement. Based on the currently used clinical definitions, CKD can be defined in patients with eGFR above this threshold in the presence of increased levels of ACR.

Thank you. We have corrected the text.

12. Diabetes and hypertension are not “chronic health conditions” (Page 4) but chronic diseases.

Thank you.

Reviewer #2 (Remarks to the Author):

The authors performed a meta-analysis on eGFR using two large published datasets of European ancestry individuals and of individuals from Japan, respectively, and included two additional smaller datasets of European ancestry.

Follow-up analyses included colocalization, lookup for different traits, and sex-stratified analyses. Although several new genetic associations with eGFR were revealed, a major drawback of this study is the lack of replication (except for the sex-stratified results). The alternatively conducted test for association of the index variants in kidney related traits in the UK Biobank addresses this problem only very limited because the tested endpoints differ from eGFR (including continuous traits vs. case/control status, renal failure vs. whole eGFR

spectrum).

We agree that replication would be important to the field going forward and we have entered discussions with Andrew Morris, Christian Pattaro and Christian Fuchsberger to enter into a joint meta-analysis with these groups in the future. However, the work we present also highlights new results from 96,329 newly analyzed individuals and a new focus on sex-specific effects. We have also attempted to replicate the sex-specific eGFR findings in sex-specific CKD estimates.

The authors state that genetic studies are needed to verify these associations in more diverse cohorts. Given that this study includes an almost equally sized sample set of EA individuals and individuals from Japan, it would be interesting to see the overlap and the differences in the eGFR associations between BBJ and the European ancestry studies which would be first step in this direction.

Thank you for this suggestion and we have added a short discussion of the differences between the Japanese and European ancestry results.

I have some additional comments and questions that should be addressed:

- Kidney-Specific eQTL Associations: As far as I understood, the significant eQTLs were constrained for not having eQTL in other tissue than kidney. The rationale for this approach is not quite clear. Variants that may have eQTLs across different tissues including kidney and would be excluded by this approach. Although this was a kidney specific eQTL analysis, it would be interesting to see also the results without removing loci that have eQTLs in non-kidney tissues.

Thank you. We have now identified eQTLs that are present in kidney tissue, and not necessarily specific to kidney tissue.

- for the DEPICT results a Bonferroni correction was applied for both tissue and geneset enrichment, which is very conservative given that in Tables S8 and S9 a FDR is provided. Thus, the authors report only the tip of the iceberg. A commonly used FDR cutoff of 5% would be more informative while controlling the type I error. Pattaro et al. (2016) reported already tissue enrichment beyond the urogenital tract that at least partially overlap with the results of this study using the proposed FDR cutoff. The authors should be consistent regarding the naming of the reported results: according to Supplementary Table 8 they report a mix of tissue names and MeSH terms of different levels for the DEPICT results, i.e. both significant tissue enrichments belong to the urogenital system.

We have now added more of the results by relaxing the threshold for inclusion to FDR 5%.

- prioritization in gene annotation: It seems not obvious that a missense variant in moderate LD ($r^2 > 0.3$) should be given a priority for gene selection. If the missense is really causal, I would expect a high LD with the lead variant, or the missense could be an independent variant which could be validated by a conditional analysis. I suggest the authors justify this criteria or increase the LD cut-off substantially. The prioritization by colocalization seems misleadingly named as eQTL in Suppl Figure 7. Finally, please provide the final selection criteria that was used to define a prioritized gene e.g. in Supplementary Table 1 and for the results of Table 2

Missense variants may not always be in high LD with the index variant if the frequencies are different (for example, R46L in PCSK9 has a frequency of 2.3% and is in r^2 of 0.03 with the initial lead variant at that locus:rs11206510 (Willer et al. 2008). However, we agree that most of the time, the 'causal' variant for an association signal will be either the lead SNP or in high LD with the lead SNP. We have removed Supplementary Figure 7, and have updated the Supplementary tables with the selection criteria for all prioritized genes. We now require the missense variant to be in $r^2 > 0.8$ with the lead variant to prioritize a gene.

- colocalization analyses of the lookup of traits in the UK Biobank: Which tissue was used for the colocalization analyses? Were the genome-wide or nominally significant associations in UK Biobank used as basis for the colocalization analysis? Did the colocalized genes per locus overlap with the eGFR colocalization results?

We only perform colocalization analysis for phenotype associations that reached our PheWAS significance threshold (requiring at least one variant in the ± 500 kb region around the eGFR index variant to have $p < 5 \times 10^{-8}$ for the tested UK Biobank trait). We are not sure how to interpret the reviewer's last sentence and wonder if the reviewer is asking whether the prioritized genes at each locus overlap with the gene identified by eQTL colocalization results? Since eQTL and DEPICT (which uses eQTL) was used to prioritize genes, the gene identification would not be independent.

- The GRS lookup with CKD was performed in the UK Biobank sample. How was CKD status defined in this UK Biobank sample? How many CKD cases and controls were available? Please provide the ROC curves of the reported AUCs as Supplemental Figures (at least for the best performing GRS).

CKD status was defined using PheCodes. We have added methods for this to the manuscript. ROC curves are now supplemental figures as well as % with CKD for each PRS bin.

- lookup of significant hits in the UK Biobank for kidney-related phenotypes (Supplementary Table 5): Please provide the effect estimates in addition to the p-values. How many cases and controls for each UK Biobank trait were included in the analyses? Were the effect directions for the CKD associations in line with the ones from eGFR? Were the effect directions for hypertension as expected with respect to eGFR? Scatterplots of the effect sizes of the eGFR

associations vs. selected UK Biobank traits would be informative.

We have added the number of cases and controls to Supplementary Table 10, and the direction for eGFR, CKD beta and se to Supplementary Table 5. We have added the scatter plots of eGFR effect size relative to hypertension effect and CKD effect as Supplementary Figure 3.

- MGI study methods: Please provide information about the time range of multiple creatinine measurements per individual that were used to average the analyzed eGFR. Furthermore, were unrelated EA individuals used (indicated in the reference publication)? Please add this information to the methods.

Added.

- sex-specific associations in the HUNT study: please specify which of the genes were prioritized as functional candidates based on colocalization and which ones on missense on high LD. Please provide also the effects of the lookup in UK Biobank in addition to the reported p-values.

We have clarified the source of evidence for biological candidate genes at each locus in new supplementary table 2.

Furthermore, the statement in the Methods "loci were identified separately in men and women and were filtered to those that were significant in one sex but near nominal significance in the other" is imprecise: which p-value exactly was defined as "near nominal"?

We have clarified this in the text ($P > 0.005$)

- Novel loci revealed in HUNT: The two reported novel loci for CKD are not plausible based on the very low MAF in combination with the number of cases, and without replication performed. A suitable MAF or minor allele count filter should be applied to the meta-analysis results, esp. for CKD. It should be mentioned, that the identified rare stop-gain variant in the known locus PKD2 still needs to be replicated in independent studies.

We have added minor allele counts and frequency within cases and controls to Supplementary Table 14 and have updated the text to point out the need for replication of these variants.

- Please provide gene names in Supplementary Tables 6, 11, 13. In addition, in Supplementary Table 13 it would be of interest to see whether prioritized genes of the conditionally independent signals differ from the prioritized genes of the main signals

We have added this information to the tables.

- Supplementary Table 1: please provide the in which order the datasets are listed in the "direction" column

Added.

- Discussion: The association of ABO with type 2 diabetes is rather weak compared to other phenotypes, therefore the postulated link of the ABO gene to CKD via type 2 diabetes is very speculative. Furthermore, given the design of this study (i.e. limiting the lookup in the UK Biobank to cardiovascular traits) and the small overlap of 6 out of 147 loci that colocalize with hypertension, I am not convinced that this study really "highlights" the pleiotropic associations with cardiovascular disease as stated. Therefore, this statement should be toned down.

Thank you, we have made this revision.

- Where any known hits not significant in the current meta-analysis analysis?

Two (of 127) previously reported variants were not significant in the current meta-analysis. These are given in supplementary table 6.

- limitations of this study should be stated, such as mix of population-based cohorts and disease cohorts (i.e. BBJ); no replication; heterogeneity in phenotype generation (different eGFR formulas and trait transformations)

Thank you for this suggestion.

- In addition to the lambdas provided in the QQ plots, LD score regression intercepts should be reported for the meta-analyses to disentangle possible inflation from polygenecity.

These are provided.

- Suppl Fig 6: There is a branch of p-values that deviates from the median line, i.e. where the p-value based meta-analysis p-value is higher than in the standard-error approach. Can the authors explain this deviation, i.e. why only a subset of associations is affected by this?

This is a set of SNPs from a locus in which the frequency differs between Japanese and European individuals, thus the square root of the sample size is not as effective at estimating the standard error when the allele frequency differs.

- Suppl Fig 6: Please rephrase "overwhelmingly" in the figure description, as a high correlation is rather expected for these type of analyses

Thank you for the careful review.

Reviewer #3 (Remarks to the Author):

The paper reports results from GWAS meta-analyses of eGFR that includes the HUNT study and MGI biobank, and two other GWAS/consortium data. They identified 147 loci including 53 novel loci. They also report sex-specific findings and associations of SNPs with multiple outcomes. There are several issues that need further clarifications. These include the strategy used for meta-analyses (minimum allele frequency, minimum number of samples contributing data), which can affect the number of identified associations, and the strategy for gene prioritization, which can affect downstream analyses. Sex-specific analyses should include interaction methods. Replication was performed using different phenotypes than those used for the discovery. Additional comments are shown below.

Introduction

2nd paragraph: CKD associations with lipids are inconsistent in studies and the statement needs to be supported by additional references.

We have added additional references in support of this.

3rd paragraph: the authors make an argument that HRC dense imputation enables discovery of new loci but a comparison of their results with imputed 1000 genomes reference panel is not included.

We have instead provided a reference that HRC imputation covers more of the genome than 1000 genomes. We feel that the effort involved in using and comparing to an older and less dense imputation reference panel is not the best science.

Results

Page 4, lines 97-98. The rationale for prioritization of genes for downstream analyses using the genes nearest to the index variant is not supported by functional evidence or the own authors' results shown in Table S4.

We only use the nearest gene if there is no other biological candidate available. Evidence suggests the nearest gene is the causal gene in ~50% of the loci.

Page 5, lines 105-108. These results are not replication as the phenotypes are disease-related and not eGFR.

We have modified the language to indicate we were examining whether variants involved in eGFR, a quantifiable measure of kidney function, are involved in chronic kidney disease. Although not a strict replication, this lends validity to the study and the eGFR association results reported.

Lines 111-114, this replication is not valid given the overlap in samples with published studies.

We have reported the association results for the current meta-analysis, which partially overlaps with previously reported studies, and for meta-analysis of HUNT and MGI alone which does not have overlap.

Lines 121-122. Specify if the eQTL is from normal kidney tissue or individuals with CKD, and if human or mouse tissue.

We have added some text to clarify these experimental details.

Page 6, lines 146-147, more details on the samples used for colocalization analysis including ancestry are needed.

We have added details to this section of the methods

Lines 155-172. How much of the eGFR heritability and variance are explained by the SNPs? The results suggest very little overall effects.

We have added an explanation of the heritability explained and how the loci can provide insight into biological underpinnings of disease, even if heritability or effect sizes are low. For example, variants in PCSK9 explain a small proportion of heritability of lipid phenotypes, but PCSK9-inhibitors have a dramatic effect on LDL cholesterol.

Line 168 statement related to differing lifestyle or hormonal factors needs to be supported by references.

We have added a reference for this.

Sex-specific analyses. Results may be driven by small samples within strata. A better strategy is using interaction methods.

We have also added this analysis. For the sex-specific loci, the variants were relatively common so the smallest stratum had between 6,895 and 7,911 individuals.

Page 7, lines 188-197, report sex-specific direction of effect for variants. Given different strategies for phenotype transformation across multiple studies, how this was compared to HUNT?.

P-value based meta-analysis was performed to look for consistently stronger associations in one sex than another. We have clarified this in the manuscript.

Lines 199-208. Should you used a more stringent p-value threshold given multiple traits tested?

We have used the standard genome-wide significance threshold of $p < 5 \times 10^{-8}$ for these analyses. We did not further adjust this threshold as each trait was analyzed separately.

Methods

Page 9, lines 263-264. HUNT study description. Is the study a longitudinal cohort or a multiple cross-sectional study?. How many individuals were recruited and how many were genotyped? Which criteria were used for selection of people for genotyping.

This is described in other publications and we have added the references. We have also added some details to the methods.

Line 280, adjustments for batch suggest that there were multiple genotyping and potential different studies combined within the HUNT study. Please clarify.

Because 70,000 samples from the HUNT study were genotyped, they were not genotyped in one enormous batch due to reagent batches etc.. Samples were randomized between batches and analyses were performed using batch as a covariate.

Page 10, lines 314-325. For the MGI, more details are needed on the visits used for serum creatinine, number of measures and timeframe selected for eGFR measures. In addition, more details are needed on relatedness and how this was accounted in the analyses.

We have added these details to the methods section.

Lines 327-330. There were also differences in eGFR equation estimation.

We have added a note of this difference to the discussion section.

Page 11, lines 334. The strategy for meta-analyses used results from at least 2 studies, so samples may be small for some of the findings, particularly for the variant with a MAF=0.01 (rs72683923). Please include the N for each variant in Table 1. Also highlight the prioritized strategy for genes for each SNP in this Table by including a column with the evidence. What was the filter for allele frequency in analyses?

We have edited supplementary table 1 to add this additional information and have clarified the methods section.

Line 343. The LD strategy for gene annotation and most significant variant within 1 Mb window are different, please clarify.

We used an LD r^2 cutoff of 0.2 for a pairwise comparison of the most significant index variants to ensure that regions with LD extending greater than 1 Mb were not mistakenly considered to be new loci. In annotating missense variants, we used an LD threshold of $r^2 > 0.8$ to identify all variants that may be in moderate LD with the index variant. We have included this in the methods section.

Line 355-258, include number of samples for each resource and if human or not. "Kidney association results" is a confusing term, please change to "eGFR associations".

Thank you, we have changed this term.

Page 12, lines 375-376, the studies chosen have differences in eGFR estimating equation that can affect the betas.

We have meta-analyzed only those studies which were inverse-normalized, so that betas are on a standardized scale.

Figure 2, the number of SNPs shown in the figure are less than 127 described in the legend, please clarify.

We have edited the figure legend to reflect that only variants with additional significant associations are plotted.

Table 1. Please include European and East Asian allele frequencies.

We have added these to supplementary table 1

Supplementary figure 3. QQ plots, the sex-specific lambda results are the same to the 6th decimal, please revise.

We have repeated this calculation and the values are correct as given. This is due to both analyses having the same median p-value.

Supplementary figures 4 and 5. Include lambdas.

We have added these

Results from supplementary figure 6 need to be included in the discussion and described in methods. Are the reported results based on p-values or standard deviations?

We have included a discussion of supplementary Figure 6 in the methods section, and have added a discussion of variants with differing p-values between the meta-analysis methods to the figure legend. The reported results are based on p-values.

Supplementary figure 7 is not needed.

We have removed this figure and have edited the supplementary tables to instead show the evidence for the gene prioritization.

Table S7. Include the source for eQTL. It looks like most are from GTEx tissues. Where are the results for the NephQTL and Ko et al.?

We have edited the supplementary tables to show the source of each tissue for the eQTL analysis.

Table S10. This information should be included in methods instead of a table.

As we have extended this table to include sample sizes and ICD-codes, we think it would be too lengthy to include in the methods section directly.